# SAEmnesia: Erasing Concepts in Diffusion Models with Supervised Sparse Autoencoders

Enrico Cassano [* 1]   Riccardo Renzulli [* 1]   Marco Nurisso [2]   Mirko Zaffaroni [3]   Alan Perotti [3]   Marco Grangetto [1]

## Abstract

Concept unlearning in diffusion models is hampered by feature splitting, where concepts are distributed across many latent features, making their removal challenging and computationally expensive. We introduce SAEmnesia, a supervised sparse autoencoder framework that overcomes this by enforcing one-to-one concept-neuron mappings. By systematically labeling concepts during training, our method achieves feature centralization, binding each concept to a single, interpretable neuron. This enables highly targeted and efficient concept erasure. Compared to the state-of-the-art sparse autoencoder-based unlearning approach, SAEmnesia reduces hyperparameter search by 96.67% and achieves a 9.22% improvement on the UnlearnCanvas benchmark for objects. Our method also shows superior scalability in sequential unlearning, improving accuracy by 28.4% when removing nine objects, establishing a step forward for precise and controllable concept erasure. Moreover, SAEmnesia effectively suppresses nudity on the I2P benchmark and remains robust to adversarial attacks. Source code available at https://github.com/EIDOS LAB/SAEmnesia

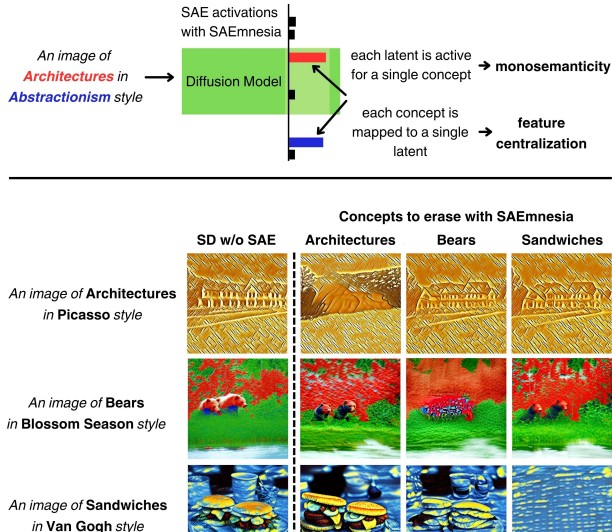

Figure 1. **SAEmnesia enables precise concept-level manipulation**: each latent activates for a single concept (*monosemanticity*), and each concept is embedded in a single latent (*feature centralization*). So, to erase a target concept, we only need to steer a single latent. The removed concepts correctly disappear in the diagonal images ("Architectures", "Bears", "Sandwiches") while the corresponding style is preserved. Note that they remain present in the non-diagonal ones, thereby preserving the fidelity and diversity when unlearning unrelated content.

## 1. Introduction

Text-to-image diffusion models have achieved remarkable success in generating high-quality images from textual descriptions, with applications across diverse domains (Rombach et al., 2022). However, they can also produce harmful, inappropriate, or copyrighted content raising safety concerns. This has spurred growing interest in machine unlearning, which aims to selectively remove undesired concepts from trained models while preserving their generative abilities (Zhang et al., 2024b).

A core challenge in concept unlearning is identifying where and how concepts are represented inside these models. Each neuron can encode multiple unrelated concepts simultaneously. This phenomenon is known as *polysemanticity*, making interpretability even more challenging. Mechanistic interpretability (MI) seeks to understand the internal workings of neural networks by analyzing their representations. Sparse Autoencoders (SAEs) provide a particularly effective MI tool by decomposing model activations into sparse and interpretable concept-level features (Bricken et al., 2023). In this work, we adopt the notion of features as the fundamental units of neural network representations that cannot

*Equal contribution  [1]University of Turin, Italy [2]Politecnico di Torino, Italy [3]Intesa Sanpaolo AI Research, Italy. Correspondence to: Enrico Cassano <enrico.cassano@unito.it>, Riccardo Renzulli <riccardo.renzulli@unito.it>.

*Proceedings of the $43^{rd}$ International Conference on Machine Learning*, Seoul, South Korea. PMLR 306, 2026. Copyright 2026 by the author(s).

be further decomposed into simpler independent factors, as defined by Bereska & Gavves (2024). Neural networks can capture natural abstractions (Chan et al., 2023) through their learned features, which serve as building blocks of their internal representations, aiming to capture the concepts underlying the data. For simplicity, we use the terms "concepts" and "features" interchangeably, as well as "neurons" and "latents". SAEs aim to learn *monosemantic* latents, meaning that they activate almost exclusively for a specific concept. On the other hand, to improve models' interpretability even further, individual concepts should not be spread across many latents, aiming for a one-to-one mapping. Yet, in practice, multiple latents often respond to the same concept: this phenomenon is also known as *feature splitting* (Bricken et al., 2023). This means modifying one concept requires changing multiple neurons. This distributed nature poses two key challenges: (i) exhaustive searches across many latent combinations to find the right subset to modify, making unlearning computationally expensive; and (ii) overlapping latents blur concept boundaries, so interventions risk unintended side effects on related concepts.

To overcome feature splitting, we introduce *SAEmnesia*, that enriches the SAE training framework enforcing one-to-one mappings between concepts and latents through supervised labeling. Therefore, our method achieves *feature centralization*, localizing each concept into a single latent and thereby preventing splitting across multiple neurons. This binding simplifies mechanistic unlearning, thanks to a precise single-latent intervention. Figure 1 illustrates the effect of SAEmnesia for concept removal in generative models, demonstrating the possibility of precise control over what the model generates, while retaining overall quality and diversity. On the UnlearnCanvas benchmark (Zhang et al., 2024b), SAEmnesia achieves a 9.22% improvement on objects over the state-of-the-art mechanistic approach (Cywiński & Deja, 2025). In sequential unlearning tasks, we demonstrate superior scalability with a 28.4% improvement in unlearning accuracy for 9-object removal. Furthermore, we reduce hyperparameter search by 96.67%. SAEmnesia also achieves overall better average results (94.85%) when both object and style unlearning are considered.

In summary, our key contributions are as follows: (i) we introduce a supervised SAE that explicitly enforces one-to-one concept–latent mappings, eliminating feature splitting and enabling transparent, interpretable control over concept representations; (ii) we show that this structure makes concept erasure significantly more efficient: each concept can be removed by steering a single latent, substantially reducing inference-time hyperparameter search; (iii) we achieve better performance on the UnlearnCanvas benchmark over state-of-the-art mechanistic approaches, improved sequential unlearning, stronger robustness to adversarial attacks, and more effective NSFW-content suppression.

## 2. Related Work

**Erasing concepts in diffusion models.** In the context of diffusion-based generative models, unlearning is often formulated as concept erasure (Feng et al., 2025), where the goal is to suppress the generation of specific visual or semantic concepts while preserving overall generation quality and unrelated content. This problem is particularly challenging due to the distributed and entangled nature of concept representations in large-scale diffusion models. Recent studies that fine-tune the entire model to unlearn concepts explore different solutions: ESD (Gandikota et al., 2023) and CA (Kumari et al., 2023) removes anchor concepts fine-tuning with negative guidance; EDiff (Wu et al., 2024) frames forgetting data as a constrained optimization problem; SA (Heng & Soh, 2023) replaces unwanted data distribution with surrogate distributions; (Li et al., 2025) propose dynamic mask coupled with concept- aware loss to improve multi-concept forgetting. Methods that do not rely on fine-tuning include SalUn (Fan et al., 2024) and SHS (Wu & Harandi, 2024) that select parameters to adapt through saliency maps or connection sensitivity; FMN (Zhang et al., 2024a) introduces a re-steering loss applied only to attention layers; SPM (Lyu et al., 2024) adds linear adapters to directly block unwanted content propagation; SEOT (Li et al., 2024) removes unwanted content from text embeddings; UCE (Gandikota et al., 2024) and RECE (Gong et al., 2025) adapt cross-attention weights using closed-form solutions; SLUG (Cai et al., 2025) identifies a single critical layer using metrics of layer importance and gradient alignment; ConceptPrune (Chavhan et al., 2025) facilities straightforward concept unlearning via weight pruning, while ScaPre (Deng et al., 2026) tackles sequential unlearning by combining a spectral trace regularizer with an information-coupling-based adaptive update rule to prevent inter-concept interference. In contrast to these approaches, our work leverages SAEs to achieve unlearning through interpretable feature manipulation during inference, without modifying the base model weights and providing full transparency into which specific features are being targeted for removal. Because the SAE is attached non-destructively, the process is fully reversible: detaching the SAE restores the original model exactly.

**SAEs background.** Our aim is to enable effective concept unlearning in diffusion models by selectively removing unwanted concepts while preserving generative quality. To achieve this, we decompose the high-dimensional, entangled activations from Stable Diffusion (SD) into sparse, interpretable feature directions that correspond to meaningful visual concepts. SAEs serve as the key tool for this decomposition, enabling us to map individual neurons to specific semantic concepts and subsequently intervene on them for targeted unlearning. A standard single-layer ReLU SAE (Olshausen & Field, 1997) operates on $d$-dimensional

activation vectors. Let $\mathbf{x} \in \mathbb{R}^d$ denote the input activation vector and $n$ be the latent dimension, typically set to $d$ multiplied by a positive expansion factor. The encoder and decoder are defined as (Bricken et al., 2023):

$$\mathbf{v} = \mathrm{ReLU}(\mathbf{W}_\mathrm{e}(\mathbf{x} - \mathbf{b}_\mathrm{p}) + \mathbf{b}_\mathrm{e}), \quad \hat{\mathbf{x}} = \mathbf{W}_\mathrm{d}\mathbf{v} + \mathbf{b}_\mathrm{p}, \quad (1)$$

where $\mathbf{v}$ is the sparse hidden representation, $\hat{\mathbf{x}}$ is the reconstructed input, $\mathbf{W}_\mathrm{e} \in \mathbb{R}^{n \times d}$ and $\mathbf{W}_\mathrm{d} \in \mathbb{R}^{d \times n}$ are the encoder and decoder weight matrices respectively, and $\mathbf{b}_\mathrm{p} \in \mathbb{R}^d$ and $\mathbf{b}_\mathrm{e} \in \mathbb{R}^n$ are learnable bias terms.

**TopK SAEs.** In our work, we employ TopK SAEs (Makhzani & Frey, 2013) that provide enhanced sparsity control. The TopK activation function identifies the $k$ largest pre-activations and sets all others to zero, ensuring sparsity while preserving the most significant features:

$$\mathbf{z} = \mathrm{TopK}(\mathbf{v}), \quad \hat{\mathbf{x}} = \mathbf{W}_\mathrm{d}\mathbf{z} + \mathbf{b}_\mathrm{p}. \quad (2)$$

Here, $\mathbf{v}$ represents the pre-TopK activations, while $\mathbf{z}$ represents the post-TopK sparse activations.

**Training objective.** Given a mini-batch of size $B$, the TopK SAE loss combines reconstruction error with an auxiliary loss to prevent dead latents:

$$\mathcal{L}_{\mathrm{unsupSAE}} = \frac{1}{B} \sum_{b=1}^{B} \left\| \mathbf{x}^{(b)} - \hat{\mathbf{x}}^{(b)} \right\|_2^2 + \alpha \, \mathcal{L}_{\mathrm{aux}}, \quad (3)$$

where $||\mathbf{x} - \hat{\mathbf{x}}||_2^2$ is the reconstruction error and $\mathcal{L}_{\mathrm{aux}}$ is an auxiliary loss using only the largest $k_{\mathrm{aux}}$ feature activations that have not fired on a large number of training samples (so-called dead latents). The auxiliary loss prevents dead latents from occurring and is scaled by coefficient $\alpha$.

**Interpretability of vision models.** SAEs have recently gained traction as a tool for uncovering human-interpretable structure in high-dimensional representations. Early applications focused on discriminative models, such as interpreting features in CLIP (Fry, 2024; Daujotas, 2024; Rao et al., 2024) or traditional classifiers (Szegedy et al., 2015; Gorton, 2024). More recent work has extended SAEs to vision–language settings, enabling tasks such as hallucination mitigation (Jiang et al., 2025) and interpretable report generation (Abdulaal et al., 2024). Across these domains, SAEs provide explicit concept–neuron mappings that move beyond post-hoc explanations and allow more precise control over learned representations. Unlike previous work on discriminative models, our approach uses SAEs with diffusion models to directly identify and control specific concepts.

Within diffusion models, interpretability research has primarily examined how semantic information propagates through the architecture. Studies have identified meaningful directions in UNet bottlenecks (Kwon et al., 2023; Park et al., 2023; Hahm et al., 2024; Ijishakin et al., 2024; Kim et al., 2025), analyzed cross-attention to link prompts with spatial activations (Tang et al., 2023; Basu et al., 2023; 2024), and even used intermediate text encoder states for generation (Toker et al., 2024). Although these advances shed light on the internals of the model, they generally do not enable targeted interventions. Our approach yields sparse representations that support both interpretation and controllable unlearning of specific concepts.

**SAE-Based unlearning.** Recent work has explored applying SAEs to diffusion models for concept manipulation. Kim & Ghadiyaram (2025) introduced Concept Steerers, training SAEs on text embedding representations to identify concept-specific directions before cross-attention processing. While their approach achieves effective concept manipulation, working only on text encoders can lead to suboptimal results, especially with adversarial attacks that exploit deeper model representations. Cywiński & Deja (2025) introduced SAeUron, a post-cross-attention approach using SAEs trained on diffusion model activations in an unsupervised manner. While SAeUron achieves state-of-the-art performance on UnlearnCanvas, its unsupervised training creates weak concept-latent mappings: individual concepts may be distributed across multiple latent features, which increases computational complexity to unlearn concepts.

## 3. Methodology

Unlike unsupervised SAE training methods (Bricken et al., 2023) that require post-hoc discovery of concept-relevant features, our supervised approach directly enforces concept-latent assignments during training to achieve stronger one-to-one mappings (see Figure 2). Although our approach requires supervision, the labels are directly derived from the text prompts used to generate or condition the images: i.e., the same concepts that the SAE aims to forget.

**SAEmnesia for diffusion models.** We apply SAEmnesia to diffusion models by training on activations extracted from every timestep $t$ of the denoising diffusion process. These activations are obtained from the cross-attention blocks of the diffusion model and form feature maps. Each feature map extracted at timestep $t$ is a spatially structured tensor of shape $\mathbf{F}_t \in \mathbb{R}^{h \times w \times d}$, where $h$ and $w$ denote the height and width of the feature map, and $d$ is the dimensionality of each feature vector. Each spatial position within the feature map corresponds to a patch in the input image. As a single SAE training sample, we consider an individual $d$-dimensional feature vector $\mathbf{x}$, disregarding the information about its spatial position. Therefore, from each feature map, we obtain $h \times w$ training samples. Note that our method is architecture-agnostic, meaning that it can be transferred to various text-to-image (T2I) models, provided that the layer(s) encoding the concepts of interest can be identified for applying SAEs.

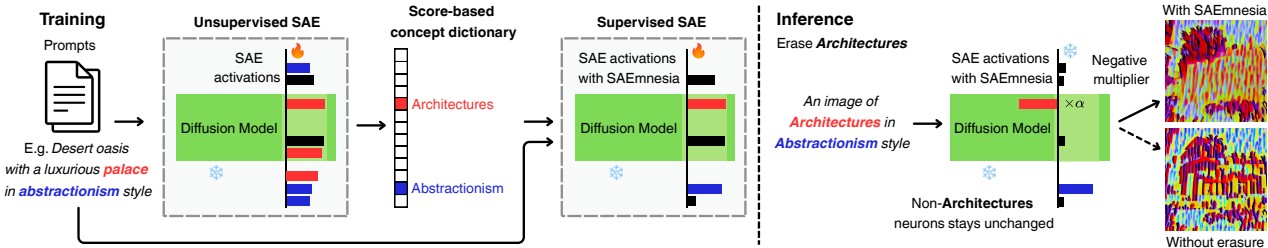

*Figure 2.* **SAEmnesia pipeline.** Training comprises two phases: (i) establishing sparse representations via standard unsupervised SAE training, (ii) applying supervised losses to strengthen specific concept-neuron associations. During inference, we need to steer a single latent per concept.

**Concept-latent assignment.** Here we introduce supervised training that directly assigns concepts to specific latents. To determine which latent should be assigned to each concept during the supervised phase and to validate the quality of these assignments after training, we utilize the score function (Cywiński & Deja, 2025), defined in Equation (4), measuring feature-concept correspondence. Given a dataset of activations $D = D_c \cup D_{\neg c}$, where $D_c$ contains data of the target concept $c$ and $D_{\neg c}$ does not, the score function is defined as:

$$\text{score}(i, t, c, D) = \frac{\mu(i, t, D_c)}{\sum_{j=1}^{n} \mu(j, t, D_c) + \delta} - \frac{\mu(i, t, D_{\neg c})}{\sum_{j=1}^{n} \mu(j, t, D_{\neg c}) + \delta} \quad (4)$$

where $\delta$ prevents division by zero and $\mu(i, t, D) = \frac{1}{|D|} \sum_{x \in D} z_i$ denotes the average activation of the $i$-th feature on activations from timestep $t$ (we omit $t$ from $z_i$ for simplicity). Features with high scores exhibit strong activation for concept $c$ while remaining weakly activated for other concepts. Formally, in order to achieve feature centralization, for a given concept $c$, the score function $\text{score}(i, t, c, D)$ achieves a high value for a single latent index $i$, while remaining low for all other indices $j \neq i$:

$$\text{score}(i, t, c, D) \gg \text{score}(j, t, c, D), \quad \forall j \neq i. \quad (5)$$

Given a set $\mathcal{C} = \{c_1, \ldots, c_K\}$ of concepts to unlearn, we define a mapping $\Phi : \mathcal{C} \to \{1, \ldots, n\}$ assigning each concept $c$ to the latent index $\Phi(c) = i_c$ with the highest score. For training samples containing multiple concepts, we assign multiple target indices corresponding to each present concept. By enforcing one-to-one concept-to-latent mappings, the cardinality of the SAE's hidden layer directly corresponds to the number of concepts that can be explicitly represented and manipulated.

**SAEmnesia loss function.** We employ a composite loss function designed to maintain SAE reconstruction capabilities while strengthening concept-latent associations:

$$\mathcal{L}_{\text{SAEmnesia}} = \mathcal{L}_{\text{unsupSAE}} + \beta \mathcal{L}_{\text{supSAE}} + \lambda \mathcal{L}_{\text{L1}}. \quad (6)$$

Where $\mathcal{L}_{\text{unsupSAE}}$ is the loss as defined in Equation (3), $\mathcal{L}_{\text{supSAE}}$ is the supervised loss that enforces the desired concept-latent relationships and $\mathcal{L}_{\text{L1}}$ is the sparsity regularization term. $\mathcal{L}_{\text{supSAE}}$ consist of our Concept Assignment (CA) loss with a weighted additional Decorrelation (DC) loss:

$$\mathcal{L}_{\text{supSAE}} = \mathcal{L}_{\text{CA}} + \eta \mathcal{L}_{\text{DC}}. \quad (7)$$

**Concept assignment loss.** Unlike traditional approaches that only apply reconstruction loss globally across all latents, our method strengthen concept-latent bonds that enable one-to-one mappings. Our CA loss is computed exclusively for latents that are assigned to concepts present in each training sample, encouraging these specific latents to activate strongly. Given a training sample, we define the binary ground-truth vector $\mathbf{y} = [y_1, \ldots, y_K]^\top \in \{0, 1\}^K$, where $y_k = 1$ if concept $c_k$ is present in the sample, and 0 otherwise. We denote with $\mathcal{T}$ the set of concepts present in that sample. The CA loss measures how much the assigned latents activate when their corresponding concepts are present:

$$\mathcal{L}_{\text{CA}} = \frac{1}{B} \sum_{b=1}^{B} \frac{1}{|\mathcal{T}^{(b)}|} \sum_{c \in \mathcal{T}^{(b)}} \left[ -\log \sigma(v_{i_c}^{(b)}) \right], \quad (8)$$

where $v_{i_c}$ represents the pre-activation (logit), and $\sigma(\cdot)$ is the sigmoid function.

**Decorrelation constraint.** To promote disentanglement across multiple macro-categories of concepts, we generalize the decorrelation constraint to $M$ disjoint concept groups. Specifically, we partition the full concept set into non-overlapping subsets $\mathcal{C} = \bigcup_{m=1}^{M} \mathcal{C}_m$, with $\mathcal{C}_m \cap \mathcal{C}_{m'} = \varnothing$ for $m \neq m'$, where each $\mathcal{C}_m$ represents a high-level group of related concepts (e.g., objects, styles, materials, or other semantic categories depending on the dataset). We denote by $\mathcal{I}_{\mathcal{C}_m} = \{\Phi(c) \mid c \in \mathcal{C}_m\}$ the set of latent indices assigned to the concepts in group $\mathcal{C}_m$. Given a mini-batch of size $B$, we define the activation vector of each concept $c$ as $\mathbf{a}_c = [v_{i_c}^{(1)}, v_{i_c}^{(2)}, \ldots, v_{i_c}^{(B)}]^\top$. The multi-group decorrela-

tion constraint is then formulated as:

$$\mathcal{L}_{\text{DC}} = \frac{\sum\limits_{m<m'} \sum\limits_{i\in\mathcal{I}_{\mathcal{C}_m}} \sum\limits_{j\in\mathcal{I}_{\mathcal{C}_{m'}}} \rho(\mathbf{a}_i, \mathbf{a}_j)}{\sum\limits_{m<m'} |\mathcal{I}_{\mathcal{C}_m}| |\mathcal{I}_{\mathcal{C}_{m'}}|}. \tag{9}$$

Here, $\rho(\mathbf{a}_i, \mathbf{a}_j)$ denotes the Pearson correlation coefficient between activation vectors $\mathbf{a}_i$ and $\mathbf{a}_j$. This constraint penalizes correlations between activation patterns of latents assigned to different concept groups.

**Sparsity regularization.** To encourage sparse activations in the latent representation, we incorporate an L1 regularization term that penalizes the magnitude of latent activations $\mathbf{v}$. This sparsity constraint promotes the emergence of interpretable features by encouraging most latents to remain inactive for any given input, thereby improving the disentanglement of learned representations.

**Feature centralization.** As defined in Equation (5), feature centralization occurs when only one feature achieves a high score for a specific concept. However, computing the score function requires the entire dataset, making it impractical during training. Instead, we use $\mathcal{L}_{\text{supSAE}}$ as a proxy to achieve feature centralization. $\mathcal{L}_{\text{CA}}$ directly enforces that assigned latents activate strongly for their target concepts, while $\mathcal{L}_{\text{L1}}$ maintains sparsity across all latents. This combination concentrates a concept's information in the designated latent.

**Inference and concept unlearning.** To unlearn a concept $c$, SAEmnesia only needs the single latent $i_c$ to erase the concept. The activation of the selected feature is multiplied by a negative value $\gamma_c < 0$ normalized by the average activation $\mu(i_c, t, D_c)$ on concept samples of a validation dataset $D$. This removes the influence of the targeted concept on the activation vector $\mathbf{z}$. Each $i_c$-th latent feature activation is modified as follows:

$$z_{i_c} = \begin{cases} \gamma_c \mu(i_c, t, D_c) z_{i_c}, & \text{if } z_{i_c} > \mu(i_c, t, D) \\ z_{i_c}, & \text{otherwise} \end{cases} \tag{10}$$

The condition $z_{i_c} > \mu(i_c, t, D)$ prevents random feature ablation when scores are low. During inference, we can use the original pretrained model for the first $t$ steps, setting the multipliers to 1 and retaining the pretrained model priors. We can then turn on SAEmnesia for the remaining steps.

## 4. Experiments and Results

We conduct comprehensive experiments to evaluate SAEmnesia across multiple dimensions: unlearning effectiveness, generation quality, concept separation, performance robustness, incremental unlearning capabilities, adversarial resilience and nudity removal. Note that our focus in this work

is on object erasure, as it presents a considerably greater challenge than style erasure (Cywiński & Deja, 2025); nevertheless, we also report style unlearning results in Appendix A.15.

### 4.1. Experimental setup

We report here the evaluation setup of SAEmnesia, including datasets, architectures and evaluation metrics.

**Dataset.** We extract activations from SD v1.5 (Rombach et al., 2022) within the UnlearnCanvas Benchmark (Zhang et al., 2024b), which contains 20 objects (e.g., "Bears", "Cats") and 50 styles (e.g., "Impressionism", "Cubism"). SD v1.x family remains the standard evaluation setting for concept-erasure research: the benchmark itself is built on SD v1.5, and most contemporary methods report results on SD v1.x (Cywiński & Deja, 2025; Thakral et al., 2025; Wang et al., 2025a;b; Lee et al., 2025; Srivatsan et al., 2025; Wu et al., 2025; Kim et al., 2026). This makes SD v1.x the appropriate and widely-accepted testbed for fair comparison and reproducibility. Nevertheless, we report preliminary experiments on SDX-Turbo in Appendix A.13.

Following SAeUron (Cywiński & Deja, 2025), we construct labeled training data by generating activations from one-sentence prompts (anchor prompts). For each object $c$, we use 80 prompts and collect feature maps from selected U-Net cross-attention blocks across all 50 denoising steps. For each style $s$, we append the postfix in ${style}$ style to the anchor prompts. Each activation is directly labeled with its corresponding object and style based on the prompt structure, creating explicit concept-activation pairs. These anchor prompts are also used for the validation dataset $D$. We focus our analysis on block up.1.1 for object-related features, as this block has been empirically demonstrated to specialize in generating specific visual aspects (Basu et al., 2023). This controlled labeling strategy provides clean supervision signals that enable direct concept-neuron mapping during SAE training, contrasting with unsupervised approaches that discover concepts through post-hoc analysis. We compute evaluation metrics on a test set with prompts An image of ${object}$ in ${style}$ style.

**SAE model.** Our best-performing method starts with a pretrained unsupervised SAE and fine-tunes it using SAEmnesia loss in Equation (6). Unless otherwise stated, SAEmnesia is applied across all denoising steps for a fair comparison with (Cywiński & Deja, 2025). For the decorrelation constraint in Equation (9), we choose $\mathcal{C}_{\text{obj}}$ and $\mathcal{C}_{\text{sty}}$, corresponding to object and style concepts, to enforce concept separation. SAE training details and SAEmnesia losses hyperparameters can be found in Appendix A.1 and A.2, respectively. Additional model setups are reported in Appendix A.8, A.9, A.10.

**Evaluation metrics.** Our primary evaluation uses the UnlearnCanvas benchmark (Zhang et al., 2024b) with Vision Transformer-based classifiers to measure three key metrics: Unlearning Accuracy (UA), which quantifies the proportion of samples from target concept prompts that are misclassified (i.e., successful unlearning); In-domain Retain Accuracy (IRA), measuring classification accuracy on retained concepts within the same domain; and Cross-domain Retain Accuracy (CRA), assessing accuracy on concepts from different domains such as object accuracy during style unlearning. To evaluate generation quality, we compute Fréchet Inception Distance (FID) scores across different unlearning configurations and multiplier values, quantifying both the quality and diversity of generated images.

### 4.2. Quantitative results

**UnlearnCanvas benchmark performance.**

Table 1 presents our performance compared to both non-SAE-based and SAE-based approaches; for SAeUron, we use the publicly available checkpoints and code released by the authors. Within the family of SAE-based unlearning methods, SAEmnesia achieves a 91.51% average score, a **+9.22** point absolute improvement over SAeUron (82.29%), with consistent gains across UA (+7.49), IRA (+5.82), and CRA (+14.34). These gains stem from the stronger concept-latent associations enabled by our supervised training, which allow more targeted interventions. While SalUn attains a higher average on object unlearning alone, SAEmnesia achieves overall better results when both object and style unlearning are considered (Table 15, Appendix).

*Table 1.* **Evaluation metrics (%) on object concept unlearning (UnlearnCanvas benchmark).** Methods are grouped by family and sorted by average score within each group. Best per metric in **bold**, second-best underlined. Within the sparse autoencoder family, SAEmnesia improves over the SAeUron baseline by +9.22 average points, with consistent gains across all metrics.

| Method | UA (↑) | IRA (↑) | CRA (↑) | Avg. (↑) |
|---|---|---|---|---|
| ESD | 92.15 | 55.78 | 44.23 | 64.05 |
| FMN | 45.64 | 90.63 | 73.46 | 69.91 |
| UCE | 94.31 | 39.35 | 34.67 | 56.11 |
| CA | 46.67 | 90.11 | 81.97 | 72.92 |
| SalUn | 86.91 | **96.35** | **99.59** | **94.28** |
| SEOT | 23.25 | 95.57 | 82.71 | 67.18 |
| SPM | 71.25 | 90.79 | 81.65 | 81.23 |
| EDiff | 86.67 | 94.03 | 48.48 | 76.39 |
| SHS | 80.73 | 81.15 | 67.99 | 76.62 |
| SAeUron | 87.16 | 85.57 | 74.14 | 82.29 |
| SAEmnesia | **94.65** | 91.39 | 88.48 | 91.51 |

We also conduct concept separation analysis by examining latent overlap between different concepts, with particular focus on the challenging "Dogs vs. Cats" classification case, a known limitation of current SAE-based unlearning

approaches (Cywiński & Deja, 2025) that leads to concept interference. Results are presented in Appendix A.11.

**Computational efficiency.** Despite introducing additional loss components during training, our method does not increase inference computational cost over SAeUron (see Appendix A.3). Furthermore, thanks to the one-to-one concept-neuron mappings, our method removes the feature combinations search complexity that affects existing methods. In fact, unsupervised SAEs require exploring different numbers of top latent features for unlearning, creating a two-dimensional search space with $m \times l$ computations, where $m = 7$ possible multiplier values and $l = 30$ number of possible latent combinations required to unlearn a concept. This results in precisely 210 evaluations. In contrast, our approach requires only $m$ computations since we only search multiplier values, as each concept maps to one neuron. This means exactly 7 evaluations are needed. This represents a 96.67% reduction in computational cost, achieved through interpretable concept localization. This efficiency makes SAEmnesia more practical for large-scale unlearning, where each concept would otherwise demand a costly latent search: with one latent per concept, only the multiplier must be tuned. Like SAeUron, our approach also naturally supports erasing multiple concepts simultaneously without any additional training. This extends to concepts absent from the SAE training set, as validated by our held-out experiments in Appendix A.15 (Table 17).

**Effect of uniform multipliers.** Figure 3 compares the effect of applying uniform multipliers (so the same multiplier for all objects) to the latent steering for SAeUron and SAEmnesia. Across all evaluation metrics, SAEmnesia consistently achieves higher and more stable scores over the full range of multipliers. This demonstrates that SAEmnesia is less sensitive to the exact steering strength across different objects and provides more robust control of concept removal. Additional quantitative plots are included in the Appendix A.12.

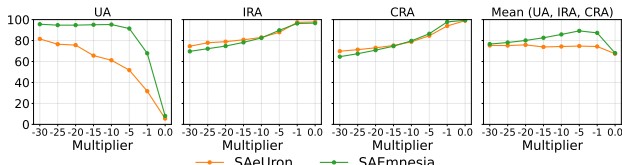

*Figure 3.* **Effect of uniform multipliers on unlearning performance for SAeUron and SAEmnesia.** SAEmnesia shows higher and more stable performance across all multipliers compared to SAeUron, indicating greater robustness to the steering strength.

**Concept-latent association distribution.** Our core contribution is creating interpretable one-to-one concept-neuron mappings. Figure 4 is an example of the effectiveness of our supervised training in promoting feature centralization by comparing feature importance score distributions before and after training for the "Flowers" concept. The original

SAeUron model exhibits relatively uniform, low-magnitude scores across the entire latent space (max score: 0.0166), indicating distributed concept representation consistent with feature splitting. In contrast, our supervised SAEmnesia model produces a clear dominant peak at neuron 11979 with a maximum score of 0.0404 (2.43 times higher than the baseline), demonstrating feature centralization. This concentrated activation pattern confirms that supervised training successfully enforces strong one-to-one concept-latent relationships, with the assigned neuron becoming highly specialized for the target concept while other neurons remain largely inactive. This transformation from distributed to concentrated representations is the foundation that enables all subsequent improvements in computational efficiency and unlearning performance. The higher score values are also tightly linked to the lower absolute values of the optimal unlearning multipliers (a deeper analysis can be found in Appendix A.5). Additional examples of score distributions are reported in the Appendix A.6.

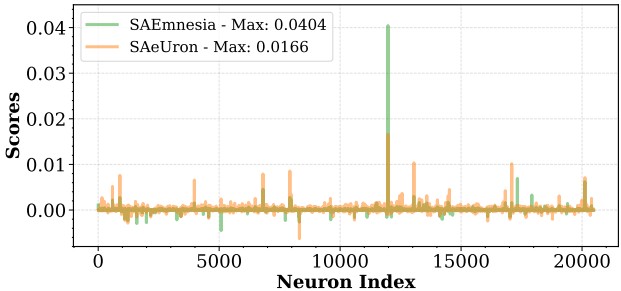

*Figure 4.* **Feature importance score distributions for "Flowers" concept.** SAeUron shows dispersed, low-magnitude scores across all neurons with a maximum of 0.0166. SAEmnesia shows a clear dominant peak at neuron 11979 with maximum score of 0.0404 (2.43× improvement).

**Feature centralization validation via K-NN classification.** To quantitatively validate that SAEmnesia achieves feature centralization, where each concept's information is concentrated in a single latent, we conducted k-nearest neighbors (k-NN) classification experiments on the latent representations across all 20 object concepts in the UnlearnCanvas benchmark. For each object, we compared classification accuracy across the denoising process using four strategies: (1) the top-scoring latent identified by SAEmnesia (Equation (4)), (2) all latent features, (3) randomly selected features, and (4) random guess baseline. As shown in Figure 5, the score-based selection achieves nearly identical classification accuracy to using all available latents throughout most of the denoising timesteps.

### 4.3. Qualitative results

**Concepts removal.** Figures 1 and 6 show qualitative examples of SAEmnesia applied to a representative subset

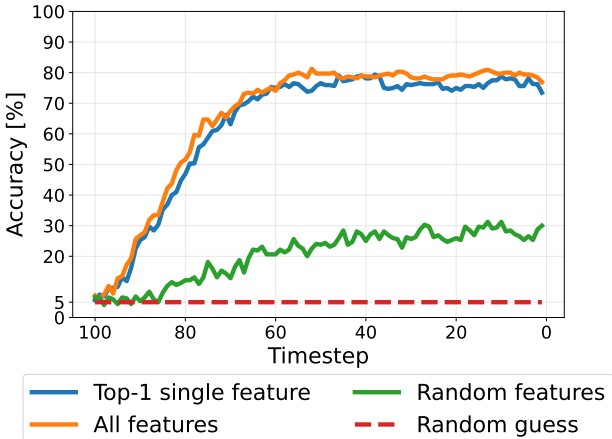

*Figure 5.* **K-NN classification across denoising timesteps, averaged over 20 object concepts.** Using only the top-scoring latent identified by SAEmnesia, performance is similar to using all features, demonstrating that supervised training successfully concentrates concept-relevant information into single interpretable latents across diverse object categories.

of object–style combinations. In this setting, we use concept multipliers of $\gamma_c = -1$ and restrict the application of SAEmnesia to the final 25 denoising steps to reduce artifacts. Thanks to the one–to–one mapping between concepts and latent units, steering the sparse autoencoder selectively removes the targeted concept: the removed concepts vanish in the diagonal images, while the corresponding style is preserved. When unlearning unrelated content, the objects in the prompts remain present in the non-diagonal ones. Additional qualitative examples covering a wider range of objects and styles are provided in Appendix A.7.

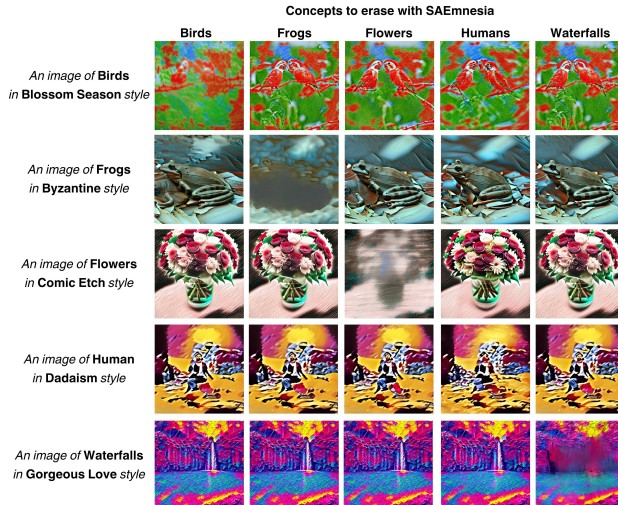

*Figure 6.* **Qualitative examples of concept removal with SAEmnesia.** Rows and columns show different styles and concepts, respectively.

**SAEmnesia effect on discovered concepts.** We examine the activation of the concept selected feature on corresponding image patches of two objects, "Architectures" and "Rabbits" across different timesteps. Figure 7 shows the difference between unsupervised SAE training and our SAEmnesia approach. The unsupervised SAE top-scoring latent (first rows) exhibits spatially incomplete and temporally inconsistent activations, suggesting that no single latent captures the full concepts, a behavior consistent with feature splitting. In contrast, SAEmnesia (second rows) produces a markedly different activation structure. This concentrated representation, where object-level semantics are encoded in a single, highly interpretable latent, enables more precise concept manipulation and unlearning through targeted interventions.

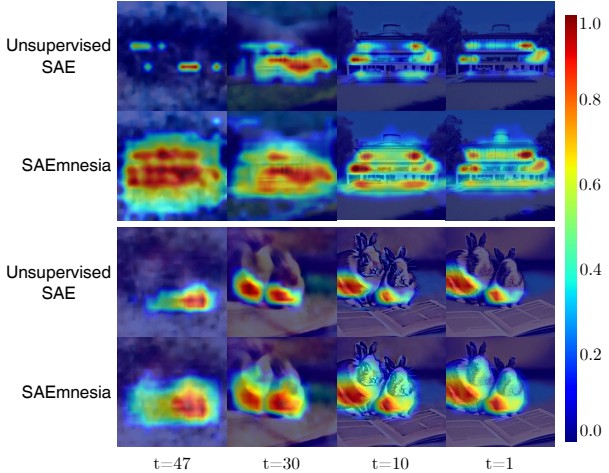

*Figure 7.* **SAEmnesia shifts attention toward patches most responsible for the target concept.** Visualization of most important patches for objects "Architectures" (top) and "Rabbits" (bottom) across different timesteps.

## 4.4. Additional experiments

**Sequential unlearning scalability.** To demonstrate how interpretable representations enable scalability, we test sequential unlearning (where unlearning requests arrive sequentially) across 9 objects: "Bears", "Cats", "Flowers", "Frogs", "Jellyfish", "Sea", "Statues", "Sandwiches" and "Waterfalls". We selected these objects because the UnlearnCanvas benchmark focuses only on the sequential unlearning of styles. In this setting, we apply the unlearning multipliers cumulatively: first $\gamma_{\text{Bears}}$ alone, then $\gamma_{\text{Bears}}$ and $\gamma_{\text{Cats}}$, and so on. Figure 8 shows that SAEmnesia significantly outperforms the baseline. SAEmnesia achieves 92.4% accuracy for 9-object removal compared to the baseline's 64%. This scalability results from having interpretable, specialized neurons rather than distributed representations that interfere with each other. SAEmnesia also achieves higher retention accuracy (RA, average of IRA and CRA), reflect-

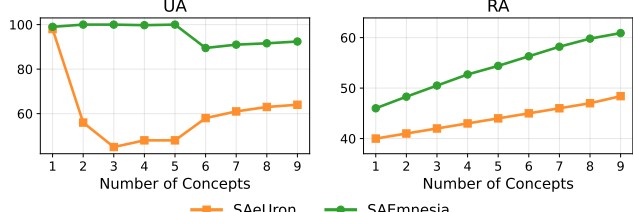

*Figure 8.* **Evaluation of SAEmnesia against SAeUron baseline on sequential object unlearning tasks.** SAEmnesia achieves higher UA and RA.

ing the model's ability to preserve all non-removed concepts. When removing all objects, SAEmnesia attains RA of 60.9%, while the baseline 48.4%.

**Adversarial robustness.** We evaluate the robustness of SAEmnesia under both white-box and black-box adversarial attacks on the UnlearnCanvas object set. Under the white-box UnlearnDiffAtk benchmark (Zhang et al., 2024c), we optimize 5-token adversarial prefixes for 40 iterations with a learning rate of 0.01. As shown in Table 2, SAEmnesia retains 57.50% unlearning accuracy under attack, compared to 34.20% for SAeUron. Relative to their clean-prompt accuracies (97.60% and 83.70% respectively), this corresponds to a 40.1-point drop for SAEmnesia versus a 49.5-point drop for SAeUron, showing that SAEmnesia is both more accurate post-attack and more stable under adversarial perturbation. To verify that this advantage is not specific to the gradient-based threat model, we further evaluate against Ring-A-Bell (Tsai et al., 2024), a black-box attack. Since the original benchmark targets NSFW and artistic-style concepts, for each object of UnlearnCanvas we generate 7 prompt pairs (containing and not containing the concept) using Claude Sonnet 4.6, extract the empirical concept vector, and run the InversePrompt genetic algorithm with the default hyperparameters. SAEmnesia attains an average Adversarial UA of 97%, against 79.50% for SAeUron, confirming that the robustness gains transfer beyond white-box settings. We hypothesize this resilience stems from the supervised concept-neuron mapping: the stronger concept-neuron bonds make it substantially harder for adversarial perturbations to disrupt the specialized representations.

*Table 2.* Object UA (%) before and after adversarial attacks, averaged over UnlearnCanvas objects.

| | UnlearnDiffAtk | | Ring-A-Bell | |
|---|---|---|---|---|
| Method | Before (↑) | After (↑) | Before (↑) | After (↑) |
| SAeUron | 83.70 | 34.20 | 82.00 | 79.50 |
| SAEmnesia | **97.60** | **57.50** | **97.50** | **97.00** |

**Nudity unlearning**. To highlight the potential of SAEmnesia in real-world applications such as NSFW content removal, we evaluate our method on the I2P bench-

mark (Schramowski et al., 2023) on SD v1.4. We follow the same experimental setting as Cywiński & Deja (2025). We train SAEs on SD-v1.4 activations gathered from a random set of 30K captions from COCO train 2014. We employ the NudeNet detector for nudity detection, filtering out outputs with confidence less than 0.6. Our best model achieves state-of-the-art results (9 detections vs. SAeUron's 18), while preserving the model's overall quality. We adopt two neurons because the SAE is trained on only two nudity-related prompts ("naked man" and "naked woman"). Complete per-category results and comparisons against all baselines are provided in Appendix A.14.

## 5. Limitations

There are several limitations of our approach serving as interesting future work directions.

**More recent diffusion models.** As mentioned in Section 4.1, while our evaluation focuses on U-Net-based DMs, a natural and well-understood testbed for concept unlearning, extending SAEmnesia to transformer-based diffusion architectures remains an important direction for future work. However, such models, including FLUX series, integrate textual and visual information differently (Gao et al., 2025) and may require architectural adaptations.

**Closed-vocabulary constraint.** When a new concept needs to be erased post-training, there are two possible approaches. The first is to train a new SAE incorporating the novel concept, which yields the strongest concept-latent binding but requires additional training. A more practical approach is to compute the score function over a small set of activations generated from prompts containing the novel concept, identify the highest-scoring latent, and use it directly for unlearning without retraining the SAE, analogously to how SAeUron handles novel concepts. This is precisely the inference-time procedure already used for concept-latent assignment, and it applies equally to unseen concepts.

**Within-group interference.** Semantically similar concepts within the same group still interfere despite the one-to-one mapping property of our method. The decorrelation constraint loss (Equation (9)) is applied to macro-groups concepts (objects vs. styles) but not within them. We note that this is still an inherent challenge in any concept unlearning method. A promising direction for future work would be to incorporate pairwise concept similarity into the supervised loss: e.g., applying stronger decorrelation penalties between highly similar concept pairs via CLIP embeddings.

**Scalability with a growing number of concepts.** While the method performs well in the current regime with 70 assigned concepts, which is comparable to or exceeding the scale of the most ambitious works in the literature, the one-concept-per-latent design may become challenging in more realistic open-vocabulary settings involving a large number of concepts (e.g., 20K+ concepts). In such cases, both latent capacity and, more importantly, stable and disentangled concept–latent assignments may become bottlenecks, particularly given observed interference between similar concepts. SAE capacity can be increased by scaling the expansion factor or latent dimension, providing a natural path to larger vocabularies. However, beyond a certain scale, unlearning a very large fraction of the model's knowledge would make full retraining on a curated dataset (if possible) a more practical and principled solution than post-hoc erasure.

## 6. Conclusions

We introduced SAEmnesia, a supervised sparse autoencoder framework for concept unlearning in diffusion models. By preventing feature splitting and enforcing concept-aligned latent structure, SAEmnesia produces more reliable concept–latent mappings and stronger erasure behavior than unsupervised SAE baselines. On UnlearnCanvas, it attains 91.51% average performance when erasing objects, while preserving competitive generation quality. The method also reduces hyperparameter search by 96.67%, substantially lowering tuning cost. In sequential unlearning, SAEmnesia scales more effectively, delivering a 28.4% improvement when removing nine objects. Looking forward, as richer benchmarks and regulatory requirements emerge, we see SAEmnesia as a step toward principled, interpretable, and controllable concept unlearning in generative models, with capabilities increasingly critical for trustworthy and safe deployment.

## Acknowledgments

We acknowledge the CINECA award under the ISCRA initiative, for the availability of high performance computing resources and support.

## Impact Statement

SAEmnesia targets concept unlearning in diffusion models based on sparse autoencoders, enabling fine-grained and interpretable control over generative behavior. By allowing the selective removal of undesirable, harmful, or copyrighted concepts without full retraining, SAEmnesia can support safe and responsible deployment of generative models, improving transparency and accountability. However, like other model editing techniques, SAEmnesia could be misused to remove safety-related behaviors or provide a false sense of security if unlearning is incomplete or unstable under distribution shifts. We therefore emphasize that this method is intended as a research tool and should be applied alongside rigorous evaluation, auditing, and governance mechanisms to ensure reliable and responsible use.

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

# A. Appendix

### A.1. SAE training details

We trained a TopK SAE with $k = 32$ and an expansion factor of 16. The model was optimized using the Adam optimizer with a learning rate of $5 \times 10^{-6}$ and a batch size of 128.

### A.2. SAEmnesia loss hyperparameters

Due to computational constraints, we selected the hyperparameters $\beta, \lambda$ and $\eta$ that achieved the lowest SAEmnesia validation loss (Equation (6)) during the first five training epochs via a grid search. We sweep three hyperparameters: $\beta \in \{1.0, 3.0, 5.0\}$, $\lambda \in \{0.005, 0.010, 0.020\}$, and $\eta \in \{0.05, 0.10, 0.20\}$, yielding 27 configurations in total. Our best experimental results were achieved with $\beta = 3.0$, $\lambda = 0.01$ and $\eta = 0.10$. Note that with higher values of $\beta$, for example when $\beta = 10$, the model produces completely white images and loses generative capabilities.

### A.3. Computational cost analysis

During training, our method adds a computational overhead compared to standard SAE training, as shown in Table 3. The overhead stems from the $\mathcal{L}_{\text{CA}}$, the $\mathcal{L}_{\text{DC}}$ and the $\mathcal{L}_{\text{L1}}$ regularization. This computational overhead is incurred only during training. Regarding the inference performance, SAEmnesia adds a 1.89% overhead to the diffusion process, compared to the 2.56% of SAeUron, as per Table 4. The value is measured across 10 images. For the experiments performed in this work, we used NVIDIA A100 GPUs with a batch size of 128.

*Table 3.* Training time performances breakdown. The values reported are in ms.

| Model | Total | $\mathcal{L}_{\text{unsupSAE}}$ | $\mathcal{L}_{\text{CA}}$ | $\mathcal{L}_{\text{DC}}$ | $\mathcal{L}_{\text{L1}}$ |
|---|---|---|---|---|---|
| SAEmnesia | $385.14 \pm 1.42$ | $71.09 \pm 0.25$ | $70.87 \pm 0.06$ | $70.89 \pm 0.08$ | $77.90 \pm 0.13$ |
| SAEURON | $71.12 \pm 0.25$ | $71.12 \pm 0.25$ | — | — | — |

*Table 4.* Overhead caused by using the SAE during inference. The values reported in the table are the seconds required to generate a single image, averaged over 10 images.

| Metric | Without SAE | SAeUron | SAEmnesia |
|---|---|---|---|
| Mean time (s) | $3.550 \pm 0.009$ | $3.641 \pm 0.011$ | $3.617 \pm 0.018$ |
| Overhead (%) | – | 2.56 | 1.89 |

### A.4. Standard deviations

We report here the standard deviations of UA, IRA and CRA metrics across 4 different seeds.

*Table 5.* Mean and standard deviations (%) of SAEmnesia on the UnlearnCanvas benchmark.

| Method | UA | IRA | CRA | Avg. |
|---|---|---|---|---|
| SAEmnesia | $94.65 \pm 2.6$ | $91.39 \pm 1.3$ | $88.48 \pm 0.5$ | $\mathbf{91.51 \pm 0.4}$ |

### A.5. Multipliers comparison

The comparison reveals significant differences in unlearning strategies between other SAE-based methods (Cywiński & Deja, 2025) and our SAEmnesia variants. SAeUron employs substantially larger multiplier magnitudes (as per Table 6), averaging -21.25, while our SAEmnesia-OS-CA-FS and SAEmnesia-OS-DC-CA-FT models achieve effective unlearning with more conservative multipliers, averaging -6.20 and -6.60 respectively. These results align well with the findings on latents score distributions shown in Figures 4 and 9a to 9c.

### A.6. Additional scores histograms

In this section are provided additional examples similar to Figure 4, to further demonstrate the robustness of our methods throughout different concepts, such as Architectures (Figure 9a), Rabbits (Figure 9b) and Sea (Figure 9c). For all the objects, we show a clear improvement, going from distributed scores (in orange) produced by SAeUron (Cywiński & Deja, 2025), to well defined peaks (in green) produced by our method SAEmnesia.

*Table 6.* Comparison of Object Unlearning Multipliers Across Three Models

| Object | SAeUron | S-OS-CA-FS | S-OS-DC-CA-FT |
|---|---|---|---|
| Architectures | -20.0 | -5.0 | -5.0 |
| Bears | -30.0 | -10.0 | -5.0 |
| Birds | -10.0 | -5.0 | -5.0 |
| Butterfly | -15.0 | -5.0 | -5.0 |
| Cats | -15.0 | -1.0 | -10.0 |
| Dogs | -20.0 | -5.0 | -5.0 |
| Fishes | -30.0 | -5.0 | -5.0 |
| Flame | -25.0 | -1.0 | -1.0 |
| Flowers | -20.0 | -5.0 | -5.0 |
| Frogs | -5.0 | -5.0 | -10.0 |
| Horses | -25.0 | -10.0 | -15.0 |
| Human | -20.0 | -5.0 | -5.0 |
| Jellyfish | -15.0 | -1.0 | -1.0 |
| Rabbits | -30.0 | -10.0 | -5.0 |
| Sandwiches | -15.0 | -25.0 | -5.0 |
| Sea | -30.0 | -5.0 | -5.0 |
| Statues | -30.0 | -1.0 | -10.0 |
| Towers | -20.0 | -5.0 | -5.0 |
| Trees | -25.0 | -5.0 | -5.0 |
| Waterfalls | -30.0 | -10.0 | -20.0 |
| **Average** | **-21.25 ± 7.45** | **-6.20 ± 5.31** | **-6.60 ± 4.48** |

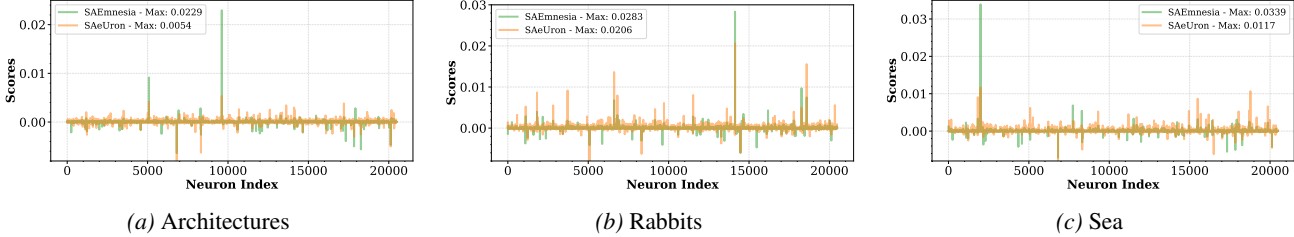

*(a)* Architectures        *(b)* Rabbits        *(c)* Sea

*Figure 9.* Feature importance score distributions across three concepts. SAeUron shows dispersed concept scores, while SAEmnesia exhibits a clear dominant peak in each case.

## A.7. Additional unlearning visualization

Figures 10 and 11 provides additional visualization with randomly sampled styles when unlearning objects with SAEmnesia. We can see that SAEmnesia performs well in most cases; however, it struggles in some scenarios with more challenging objects that are more blended with the styles, for example, flames and jellyfishes. Similarly, Figure 12 provides qualitative results for style unlearning.

Figures 13 and 14 show the unlearning accuracies for each object in the columns when unlearning the objects in the rows. This visualization highlights entanglements between different objects that can lead to poor unlearning performance.

## A.8. Global cross-entropy

We employ Cross-entropy (CE) loss with softmax applied to all pre-TopK latent activations as an alternative to the Concept-Assignment Loss (Equation (8)). This way, we treat concept assignment as a classification problem across the entire latent space:

$$\mathcal{L}_{\text{global-ce}} = -\frac{1}{B} \sum_{i=1}^{B} \log \left( \frac{\exp(v_{c_i}^i)}{\sum_{j=1}^{n} \exp(v_j^i)} \right), \tag{11}$$

where $B$ is the batch size, $v_j^i$ represents the pre-TopK activation value for sample $i$ at latent position $j$, $c_i$ denotes the assigned concept latent index for sample $i$, and $n$ is the total number of latents.

This approach, however, led to suboptimal performance as reported in Tables 8 and 9 w.r.t. OO-GCE-FS and OO-GCE-FT.

*Table 7.* SAE Experiment Versions.

| Feature/Metric | OO-GCE-FS | OO-GCE-FT | OS-DC-CA-FS | OS-DC-CA-FT | OO-CA-FS | OO-CA-FT | OS-CA-FS | OS-CA-FT | OS-TK-CA-FS | OS-TK-CA-FT |
|---|---|---|---|---|---|---|---|---|---|---|
| Object labels | ✓ | ✓ | ✓ | ✓ | ✓ | ✓ | ✓ | ✓ | ✓ | ✓ |
| Styles labels | ✗ | ✗ | ✓ | ✓ | ✗ | ✗ | ✓ | ✓ | ✓ | ✓ |
| Decorrelation | ✗ | ✗ | ✓ | ✓ | ✗ | ✗ | ✗ | ✗ | ✗ | ✗ |
| Global CE | ✓ | ✓ | ✗ | ✗ | ✗ | ✗ | ✗ | ✗ | ✗ | ✗ |
| CA before Top-K | ✗ | ✗ | ✓ | ✓ | ✓ | ✓ | ✓ | ✓ | ✗ | ✗ |
| CA after Top-K | ✗ | ✗ | ✗ | ✗ | ✗ | ✗ | ✗ | ✗ | ✓ | ✓ |
| Finetuned | ✗ | ✓ | ✗ | ✓ | ✗ | ✓ | ✗ | ✓ | ✗ | ✓ |

## A.9. Post-TopK loss analysis

OS-TK-CA-FT is the only model variant that applies CA loss after the Top-K, yet demonstrates competitive performance without major degradation. In fine-tuned configs with hyperparams search (Table 8), OS-TK-CA-FT achieves 85.52% average performance compared to baseline's 82.29%, a meaningful 3.2% improvement. Similarly, in from-scratch training (Table 9), OS-TK-CA-FT maintains 86.93% performance, indicating that post Top-K supervision remains viable.

## A.10. Exploratory variants

In Table 7, all the variants tested for this work are presented. SAEmnesia-OS-DC-CA-FT is the version reported as SAEmnesia in the main paper.

*Table 8.* SAEmnesia Experimental Results - Fine Tuned Models - Searched HP

| Metric | SAeUron | OO-GCE-FT | OS-DC-CA-FT | OO-CA-FT | OS-CA-FT | OS-TK-CA-FT |
|---|---|---|---|---|---|---|
| UA (%) ↑ | 87.16 | 94.55 | 91.75 | 95.75 | 90.90 | 66.85 |
| IRA (%) ↑ | 85.57 | 87.44 | 93.16 | 85.92 | 92.11 | 96.76 |
| CRA (%) ↑ | 74.14 | 57.71 | 88.60 | 75.09 | 86.46 | 92.95 |
| Avg. (%) ↑ | 82.29 | 79.9 | **91.51** | 85.59 | 89.82 | 85.52 |
| FID (%) ↓ | 124.11 | 155.17 | 111.16 | 111.70 | 110.84 | 110.24 |

*Table 9.* SAEmnesia Experimental Results - Trained From Scratch with Hyperparameters Search

| Metric | SAeUron | OO-GCE-FS | OS-DC-CA-FS | OO-CA-FS | OS-CA-FS | OS-TK-CA-FS |
|---|---|---|---|---|---|---|
| UA (%) ↑ | 87.16 | 95.98 | 92.95 | 78.45 | 86.00 | 69.45 |
| IRA (%) ↑ | 85.57 | 68.20 | 91.48 | 64.25 | 92.76 | 97.38 |
| CRA (%) ↑ | 74.14 | 42.48 | 88.12 | 57.76 | 87.05 | 93.97 |
| Avg. (%) ↑ | 82.29 | 69.22 | **90.85** | 66.82 | 88.93 | 86.93 |
| FID (%) ↓ | 124.11 | 150.15 | 110.25 | 119.46 | 110.49 | 109.84 |

## A.11. Concept interference mitigation

Dogs vs. Cats overlap analysis (Tables 10 and 11) provides crucial insights into concept interference patterns, a known limitation of the current approaches (Cywiński & Deja, 2025). From-scratch models generally show better concept separation, with OO-GCE-FS achieving zero overlapping timesteps. Fine-tuned ones show more variable performance, with some variants (OO-GCE-FT, OS-TK-CA-FT) achieving zero overlap while others (OS-CA-FT) perform worse than baseline.

## A.12. Uniform multipliers analysis

The uniform multiplier sweep analysis (Figures 15 and 16) reveals varying performance characteristics across model versions and evaluation metrics. In the fine-tuned models (Figure 15), SAEmnesia variants show competitive performance with some variations across different multiplier ranges, with certain variants like S-OS-DC-CA-FT maintaining strong performance while others such as S-OS-TK-CA-FT exhibit sensitivity to specific multiplier settings. The from-scratch models (Figure 16) demonstrate different behavior patterns, with variants like S-OO-CA-FS showing particular sensitivity to moderate multiplier values before recovering at gentler settings. Both figures indicate that SAEmnesia variants achieve reasonable performance across various hyperparameter ranges. The results suggest that different SAEmnesia configurations may be better suited for different multiplier ranges, highlighting the importance of hyperparameter selection in optimizing unlearning performance.

*Table 10.* Timesteps with Cats-Dogs Overlapping Latent as Most Active - Models Trained from scratch

| Metric | SAeUron | OO-GCE-FS | OS-DC-CA-FS | OO-CA-FS | OS-CA-FS | OS-TK-CA-FS |
|---|---|---|---|---|---|---|
| Overlap. timesteps | 10 | 0 | 3 | 6 | 10 | 2 |

*Table 11.* Timesteps with Cats-Dogs Overlapping Latent as Most Active - Finetuned Models

| Metric | SAeUron | OO-GCE-FT | OS-DC-CA-FT | OO-CA-FT | OS-CA-FT | OS-TK-CA-FT |
|---|---|---|---|---|---|---|
| Overlap. timesteps | 10 | 0 | 10 | 3 | 14 | 0 |

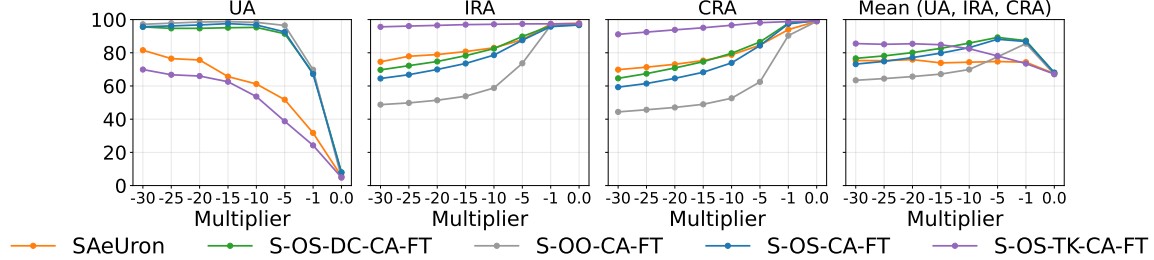

*Figure 15.* All Model's performances with uniform multipliers - fine tuned.

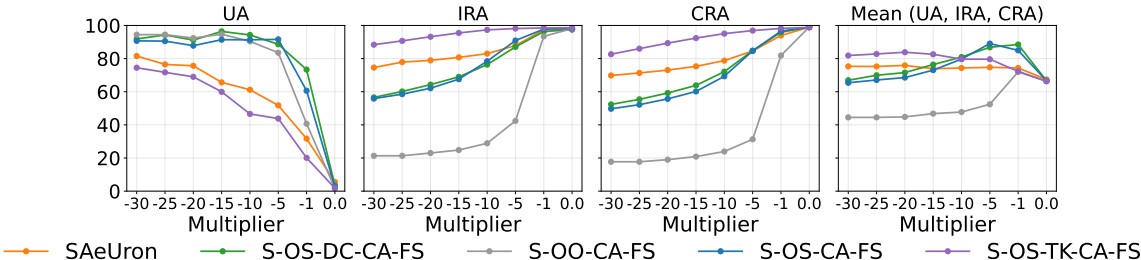

*Figure 16.* All Model's performances with uniform multipliers - from scratch.

## A.13. Additional diffusion backbones

To assess whether the supervised concept-neuron mapping generalizes beyond the SD v1.5 backbone used in our main evaluation, we report a preliminary comparison on a different backbone in Table 12. We evaluate three settings: no unlearning (a trivial baseline that simply leaves the model untouched), an unsupervised SAE (Surkov et al., 2026), and SAEmnesia.

*Table 12.* Object unlearning preliminary results SDXL Turbo.

| Method | UA (↑) | IRA (↑) | Avg. (↑) |
|---|---|---|---|
| No unlearning | 0.00 | 100.00 | 50.00 |
| Unsupervised SAE (Surkov et al., 2026) | 13.92 | **85.55** | 49.73 |
| SAEmnesia | **63.63** | 45.15 | **54.39** |

The unsupervised SAE barely exceeds the trivial baseline (49.73 vs. 50.00 Avg.), whereas SAEmnesia substantially improves UA from 13.92% to 63.63%, confirming that the supervised signal generalizes to a different backbone. The IRA drop and the omission of CRA reflect a domain gap in the evaluation protocol: UnlearnCanvas uses a fine-tuned SD v1.5 generator and ViT classifiers trained on SD v1.5 images, making direct metric comparison on a different backbone noisy. A fair evaluation on this backbone requires dedicated retraining of the evaluation classifiers, which we identify as future work.

## A.14. Nudity unlearning

For nudity unlearning, we follow the same experimental setting as Cywiński & Deja (2025) as described in the main paper. Our base unlearning setup uses only the top scoring latent to erase an object. As per Table 13, for nudity unlearning, SAEmnesia achieves weak performance (47 detections vs. SAeUron's 18 detections). However, when we instead steer two latents in our variant SAEMNESIA-TOP2, performance substantially improves (9 detections). The highly imbalanced distribution of nudity-related content in the training dataset may lead to a weaker concept centralization.

Nevertheless, SAEmnesia maintains a significant practical advantage over unsupervised alternatives. SAeUron requires steering the 205 top-scoring latents to achieve such performances, while SAEMNESIA-TOP2 only needs 2.

*Table 13.* SAEMNESIA-TOP2 is the presented SAEmnesia setup, but with two latents steered instead of one. Multiplier for SAEmnesia (one latent affected): -49. Multiplier for SAEMNESIA-TOP2 (two latents affected): -600.

| Method | Armpits | Belly | Buttocks | Feet | Breasts (F) | Genitalia (F) | Breasts (M) | Genitalia (M) | Total | CLIPScore (↑) | FID (↓) |
|---|---|---|---|---|---|---|---|---|---|---|---|
| FMN | 43 | 117 | 12 | 59 | 155 | 17 | 19 | 2 | 424 | 30.39 | 13.52 |
| CA | 153 | 180 | 45 | 66 | 298 | 22 | 67 | 7 | 838 | 31.37 | 16.25 |
| AdvUn | 8 | 0 | 0 | 13 | 1 | 1 | 0 | 0 | 28 | 28.14 | 17.18 |
| Receler | 48 | 32 | 3 | 35 | 20 | 0 | 17 | 5 | 160 | 30.49 | 15.32 |
| MACE | 17 | 19 | 2 | 39 | 16 | 0 | 9 | 7 | 111 | 29.41 | 13.42 |
| CPE | 10 | 8 | 2 | 8 | 6 | 1 | 3 | 2 | 40 | 31.19 | 13.89 |
| UCE | 29 | 62 | 7 | 29 | 35 | 5 | 11 | 4 | 182 | 30.85 | 14.07 |
| SLD-M | 47 | 72 | 3 | 21 | 39 | 1 | 26 | 3 | 212 | 30.90 | 16.34 |
| ESD-x | 59 | 73 | 12 | 39 | 100 | 6 | 18 | 8 | 315 | 30.69 | 14.41 |
| ESD-u | 32 | 30 | 2 | 19 | 27 | 3 | 8 | 2 | 123 | 30.21 | 15.10 |
| SAeUron | 7 | 1 | 3 | 2 | 4 | 0 | 0 | 1 | 18 | 30.89 | 14.37 |
| SAEmnesia | 7 | 17 | 2 | 5 | 11 | 2 | 2 | 1 | 47 | 30.98 | 14.72 |
| SAEMNESIA-TOP2 | 1 | 3 | 1 | 0 | 4 | 0 | 0 | 0 | 9 | 30.98 | 14.72 |
| SD v1.4 | 148 | 170 | 29 | 63 | 266 | 18 | 42 | 7 | 743 | 31.34 | 14.04 |
| SD v2.1 | 105 | 159 | 17 | 60 | 177 | 9 | 57 | 2 | 586 | 31.53 | 14.87 |

## A.15. Styles unlearning

We report style unlearning performance obtained with SAEmnesia in Table 14. While style erasure is generally considered an easier setting than object erasure (Cywiński & Deja, 2025), these results demonstrate that SAEmnesia remains competitive also in this regime. Notably, our method matches or slightly improves upon SAeUron across all metrics, confirming that enforcing concept-aligned sparse representations does not hinder performance on style-based unlearning tasks.

*Table 14.* SAEmnesia performances for styles unlearning.

| Method | UA | IRA | CRA | Avg. | FID (↓) |
|---|---|---|---|---|---|
| SAeUron | 95.80% | 99.10% | 99.40% | 98.10% | 1.26 |
| SAEmnesia | 96.60% | 98.67% | 99.30% | 98.19% | 1.14 |

Complete object and style unlearning results are shown in Table 15.

**Sequential style unlearning.** We additionally evaluate sequential unlearning on the UnlearnCanvas style benchmark following SAeUron, where styles are erased one at a time and accuracy is measured after each step. As reported in Table 16, both methods start at comparable performance, but SAEmnesia's accuracy increases as more styles are erased, while SAeUron remains essentially flat. This suggests that the supervised concept-neuron mapping becomes increasingly advantageous as the number of sequentially unlearned concepts grows.

**Generalization to held-out concepts.** Following the held-out protocol of SAeUron (Cywiński & Deja, 2025), we train the supervised SAE on half of the UnlearnCanvas styles and evaluate erasure on the remaining, unseen styles, directly measuring generalization beyond the supervised concept set. As reported in Table 17, SAEmnesia outperforms SAeUron on all three metrics, indicating that the supervised concept-neuron mapping does not merely overfit to the training concepts but transfers to concepts unseen during SAE training.

*Table 15.* **Evaluation of SAEmnesia against state-of-the-art methods on style and object unlearning.** Best results are in **bold** and second-best are underlined.

| Method | Effectiveness | | | | | | | Efficiency | | |
| | Style Unlearning | | | Object Unlearning | | | Avg. (↑) | FID (↓) | Memory | Storage |
| | UA (↑) | IRA (↑) | CRA (↑) | UA (↑) | IRA (↑) | CRA (↑) | | | (GB) (↓) | (GB) (↓) |
|---|---|---|---|---|---|---|---|---|---|---|
| ESD (Gandikota et al., 2023) | **98.58** | 80.97 | 93.96 | 92.15 | 55.78 | 44.23 | 77.61 | 65.55 | 17.8 | 4.3 |
| FMN (Zhang et al., 2024a) | 88.48 | 56.77 | 46.60 | 45.64 | 90.63 | 73.46 | 66.93 | 131.37 | 17.9 | 4.2 |
| UCE (Gandikota et al., 2024) | 98.40 | 60.22 | 47.71 | 94.31 | 39.35 | 34.67 | 62.45 | 182.01 | 5.1 | 1.7 |
| CA (Kumari et al., 2023) | 60.82 | 96.01 | 92.70 | 46.67 | 90.11 | 81.97 | 78.05 | **54.21** | 10.1 | 4.2 |
| SalUn (Fan et al., 2024) | 86.26 | 90.39 | 95.08 | 86.91 | **96.35** | **99.59** | 92.43 | 61.05 | 30.8 | 4.0 |
| SEOT (Li et al., 2024) | 56.90 | 94.68 | 84.31 | 23.25 | 95.57 | 82.71 | 72.91 | 62.38 | 7.34 | **0.0** |
| SPM (Lyu et al., 2024) | 60.94 | 92.39 | 84.33 | 71.25 | 90.79 | 81.65 | 80.23 | 59.79 | 6.9 | **0.0** |
| EDiff (Wu et al., 2024) | 92.42 | 73.91 | 98.93 | 86.67 | 94.03 | 48.48 | 82.41 | 81.42 | 27.8 | 4.0 |
| SHS (Wu & Harandi, 2024) | 95.84 | 80.42 | 43.27 | 80.73 | 81.15 | 67.99 | 74.90 | 119.34 | 31.2 | 4.0 |
| SAeUron (Cywiński & Deja, 2025) | 95.80 | **99.10** | **99.40** | 87.16 | 85.57 | 74.14 | 90.10 | 62.69 | **2.8** | 0.2 |
| **SAEmnesia (ours)** | 96.60 | 98.67 | 99.30 | **94.65** | 91.39 | 88.48 | **94.85** | 56.15 | **2.8** | 0.2 |

*Table 16.* Sequential style unlearning. Unlearning accuracy (UA) as a function of the number of styles sequentially erased.

| Method | Abstractionism | Byzantine | Cartoon | Cold Warm | Ukiyoe | Van Gogh |
|---|---|---|---|---|---|---|
| SAeUron | **0.9710** | **0.9718** | 0.9705 | 0.9710 | 0.9716 | 0.9722 |
| SAEmnesia | 0.9684 | 0.9698 | **0.9905** | **0.9968** | **0.9928** | **0.9904** |

*Table 17.* Held-out concept generalization. Supervised training uses half of the UnlearnCanvas styles; erasure is evaluated on the remaining (unseen) styles.

| Method | UA (↑) | IRA (↑) | CRA (↑) | Avg. (↑) |
|---|---|---|---|---|
| SAeUron | 50.69 | 68.11 | 98.16 | 72.32 |
| SAEmnesia | **53.07** | **69.93** | **98.26** | **73.75** |

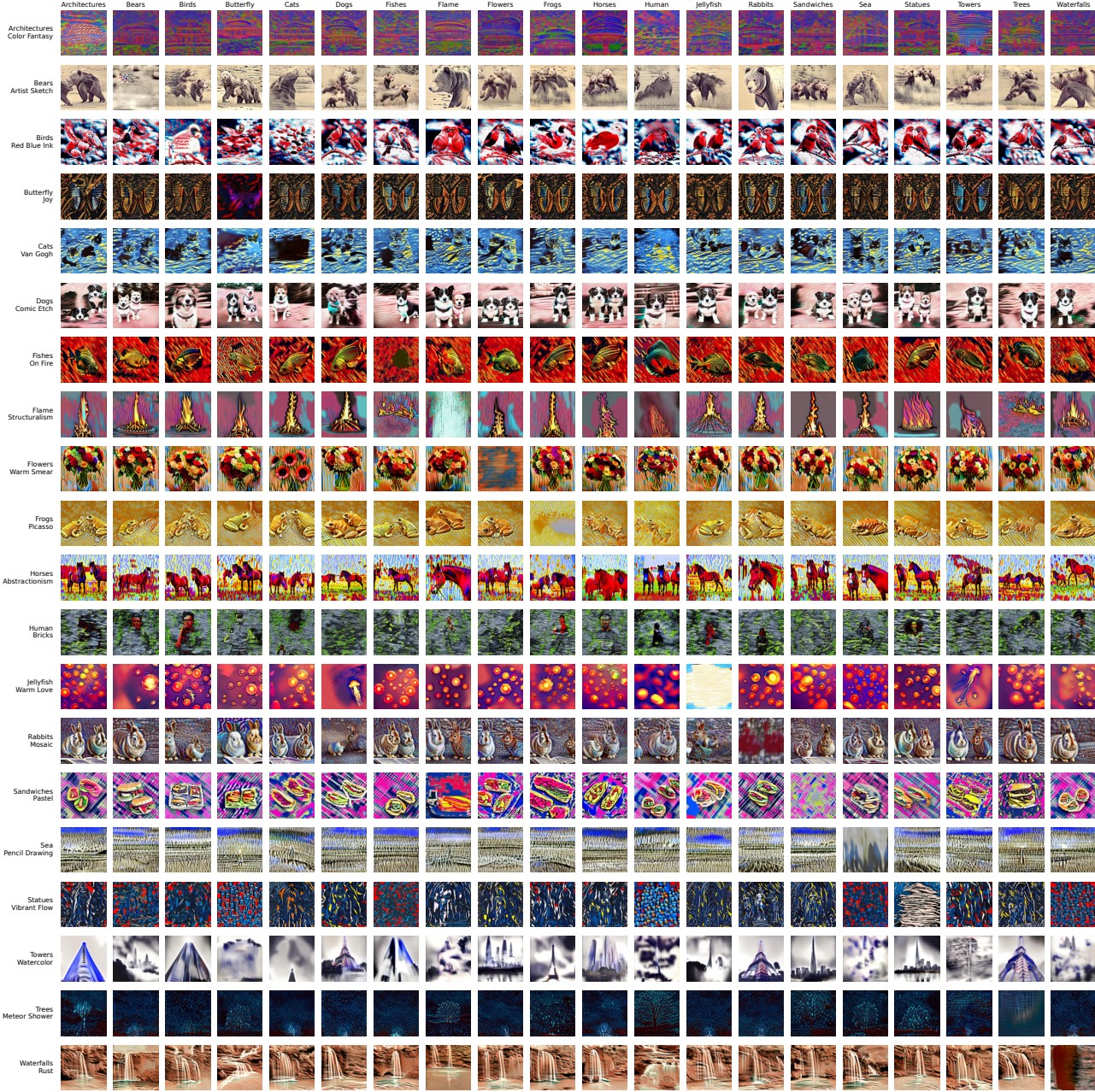

*Figure 10.* Qualitative examples of concept removal with SAEmnesia. Each row shows a different randomly sampled style, and each column a different object concept.

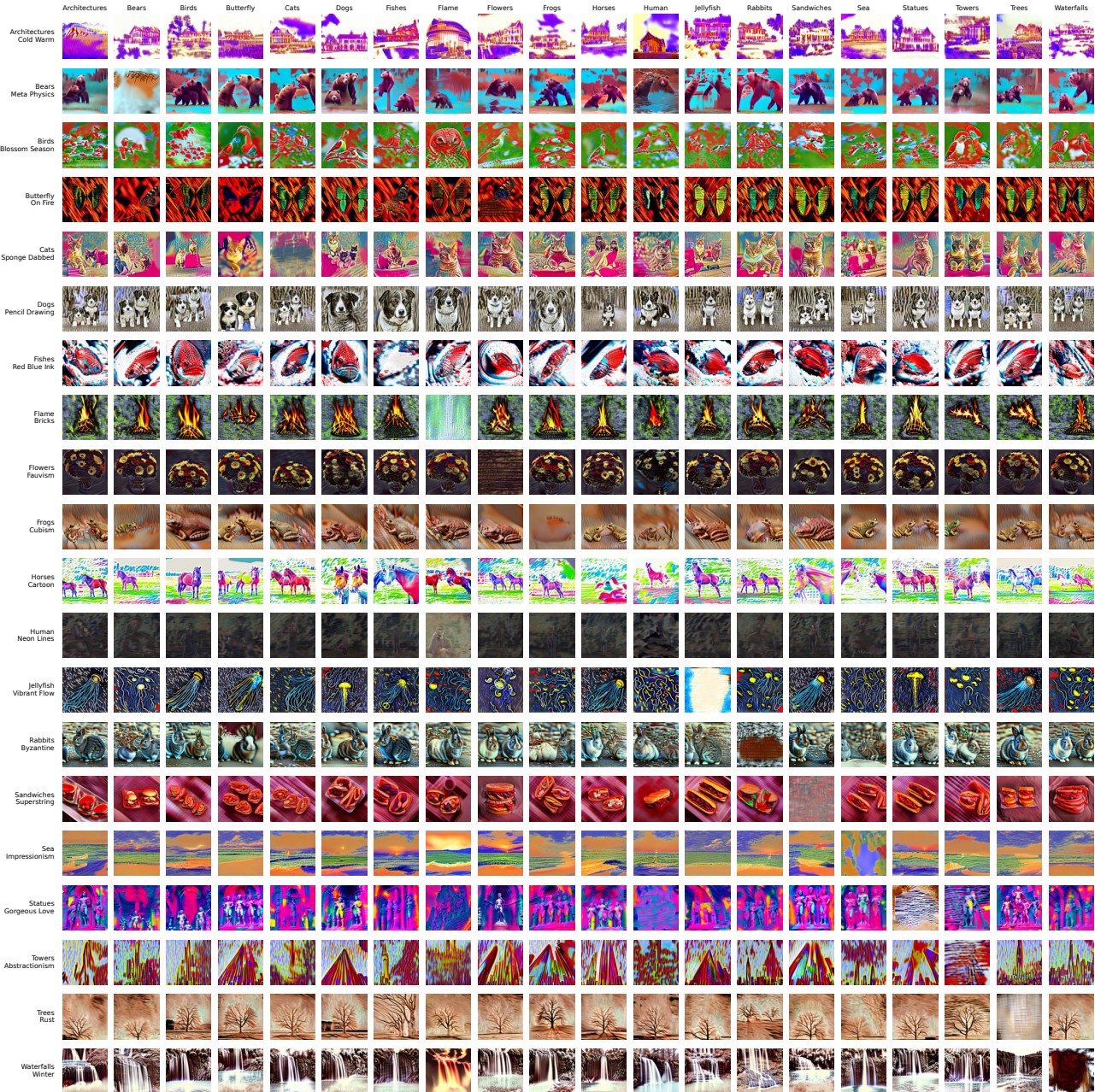

*Figure 11.* Qualitative examples of concept removal with SAEmnesia. Each row shows a different randomly sampled style, and each column a different object concept.

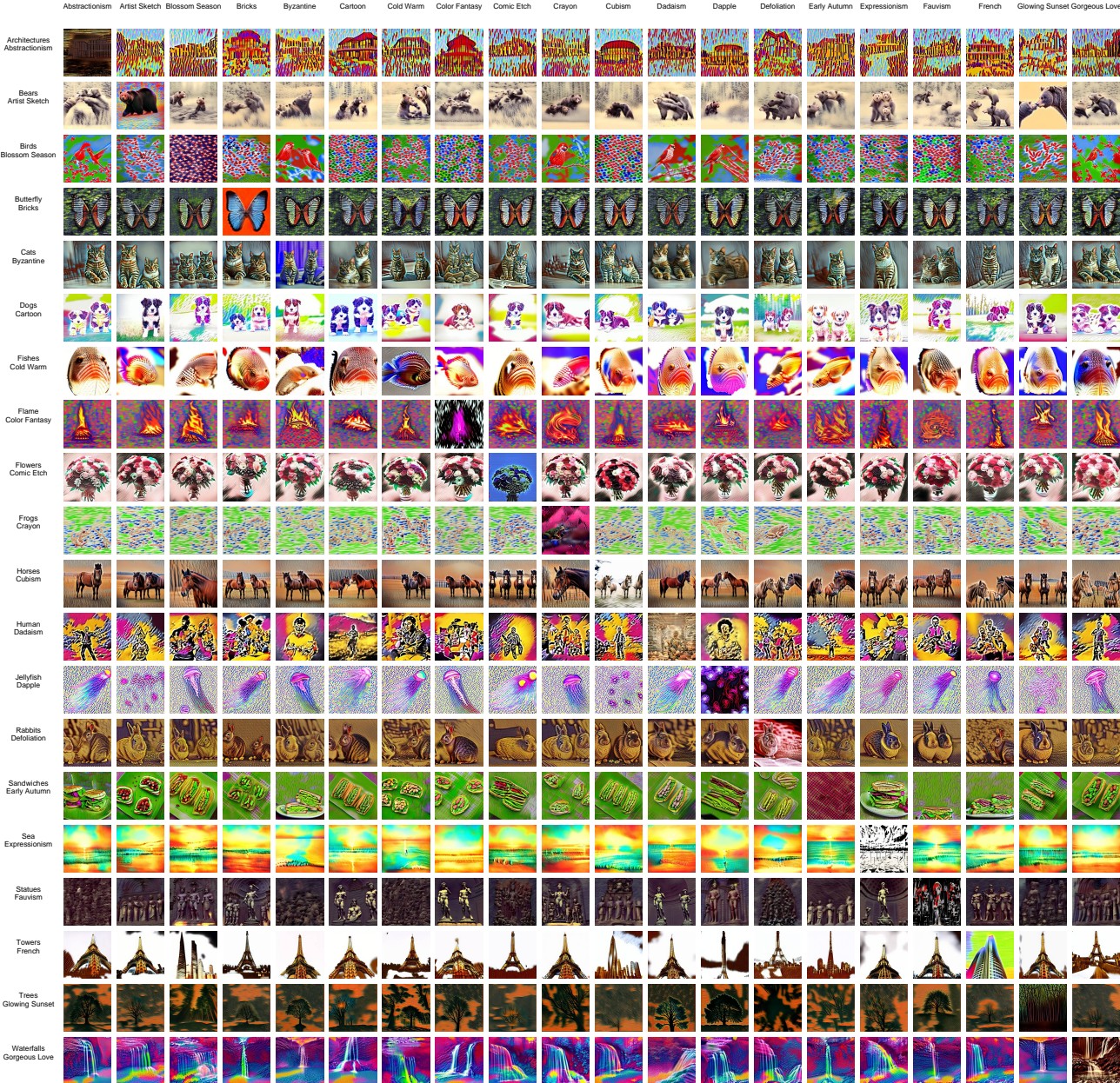

*Figure 12.* Qualitative examples of styles removal with SAEmnesia. Each row shows a different combination of object and style, and each column a different style to be unlearnt.

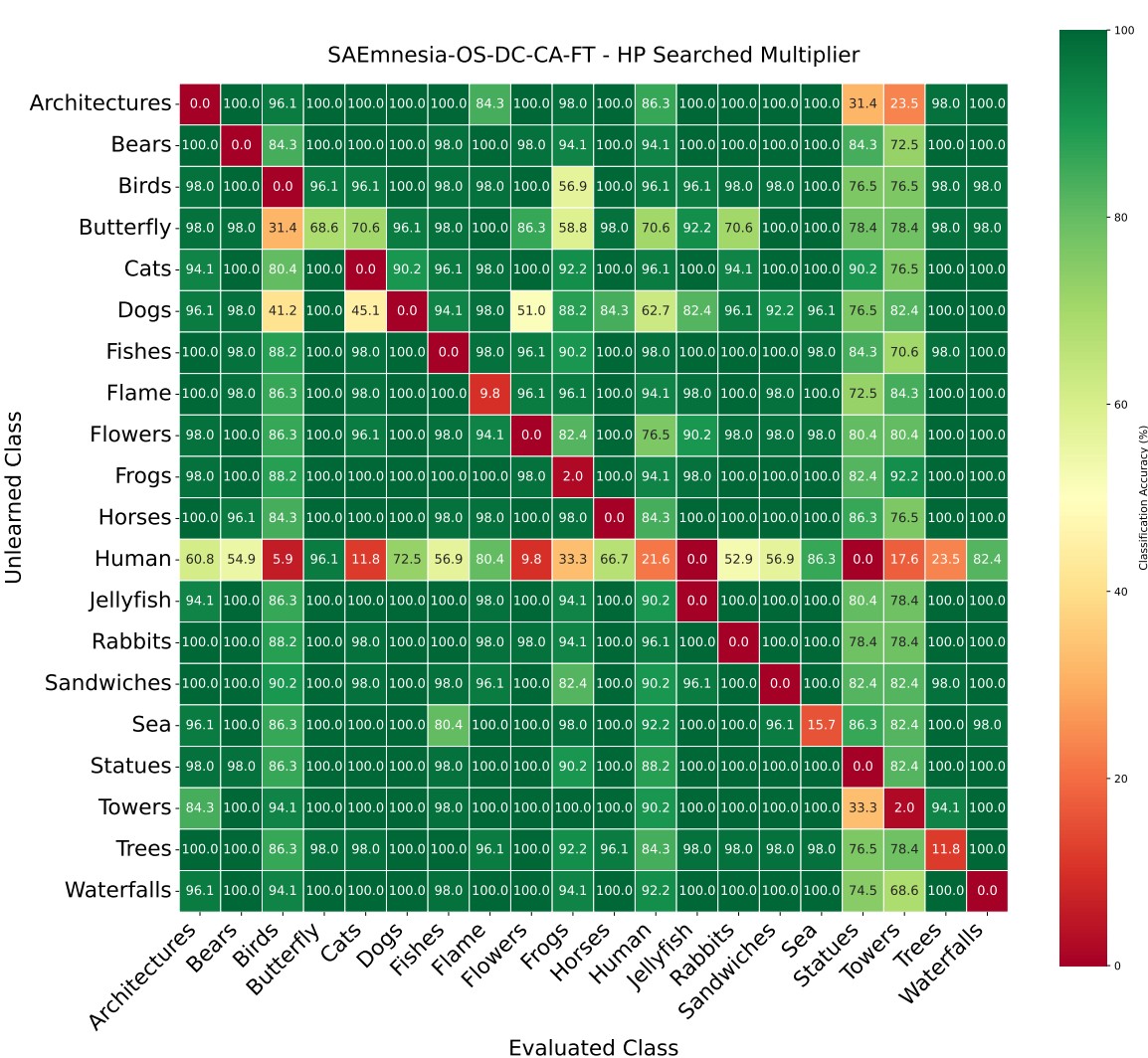

*Figure 13.* OS-DC-CA-FT UA and IRA classes disentanglement.

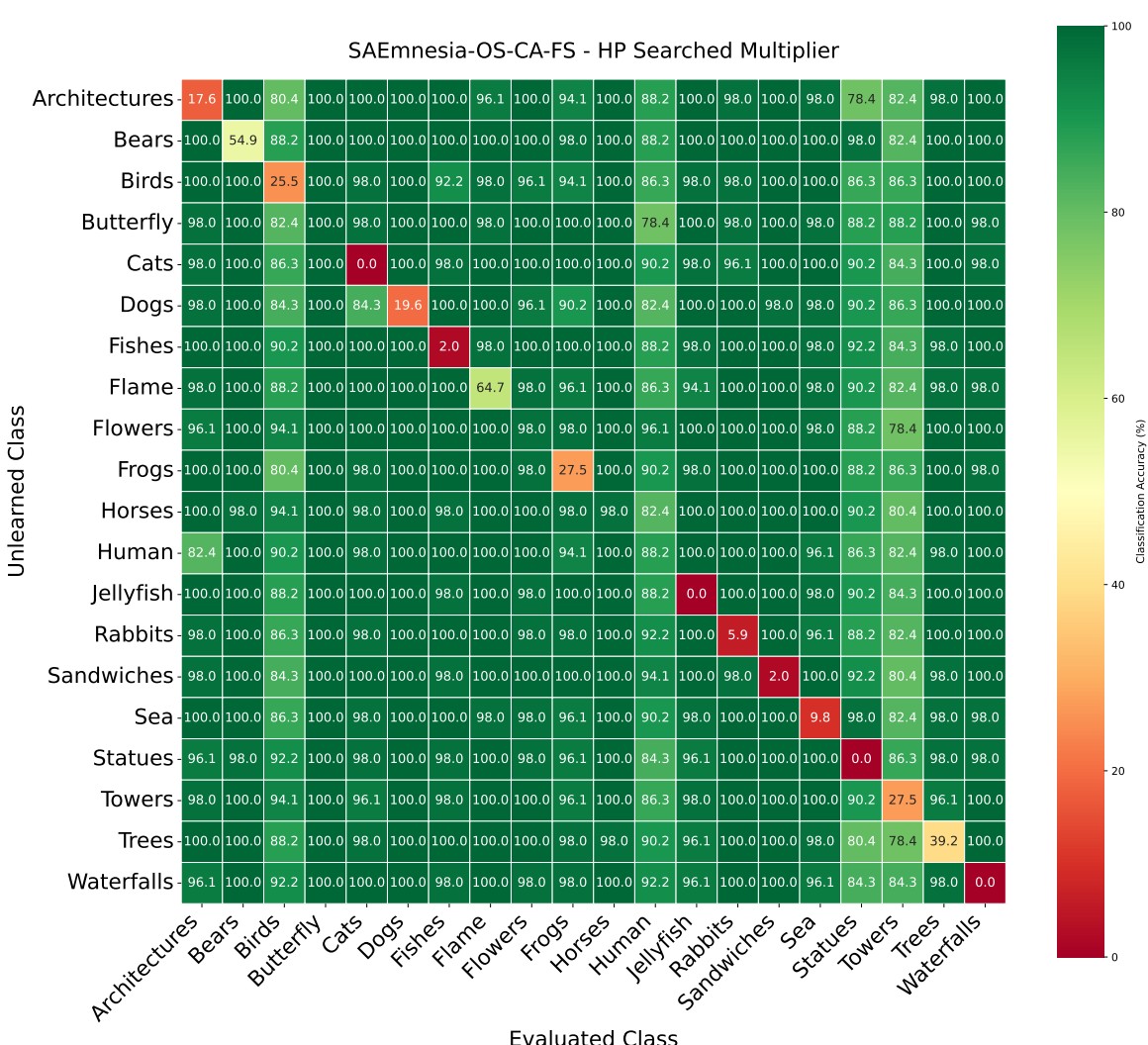

*Figure 14.* OS-CA-FS UA and IRA classes disentanglement.

