# OpenReview forum: "SAEmnesia: Erasing Concepts in Diffusion Models with Supervised Sparse Autoencoders"
_ICML.cc/2026/Conference — ICML 2026 regular_

### Official Review · Reviewer_aQBR · 2026-02-28

**Soundness:** 2
**Presentation:** 2
**Significance:** 3
**Originality:** 2
**Overall Recommendation:** 3
**Confidence:** 2

**Summary:**

The paper proposes SAEMNESIA, a supervised sparse autoencoder that improves concept unlearning in diffusion models by enforcing one-to-one concept-neuron mappings through systematic labeling during training.

**Compliance With Llm Reviewing Policy:**

Affirmed.

**Key Questions For Authors:**

As pointed out in line 151, the method applies to various text-to-image (T2I) models, so it would be more convincing to experiment with more models.

There are many hyperparameters required by the methods, such as those in Eq. 6 and 7. It is not clear how they are simultaneously optimized during the five training epochs.

**Limitations:**

Yes, the limitations have been discussed.

**Strengths And Weaknesses:**

The paper shows an improved performance better than previous methods by 9.2% on UnlearnCanvas.

In Table 1, some key models and metrics are missing, according to the official UnlearnCanvas benchmark. For example, SalUn is not compared, and Style-UA, Style-IRA, Style-CRA, and FID are not measured. Specifically, in the official UnlearnCanvas leaderboard (https://huggingface.co/spaces/OPTML-Group/UnlearnCanvas-Benchmark), SalUn appears to be a superior method (CRA = 99.59%), but it is not compared here in Table 1.

How does the method work on unlearning similar but different concepts?

It seems that in some situations the method will fail. For example, in Figure 12, the unlearning of "Dogs" and "Horses" is not very successful,

---

> ### Author Rebuttal · Authors · 2026-03-30
>
> **W1 - Missing baselines and metrics (SalUn, Style metrics, FID)**
>
> We thank the reviewer for pointing this out. Regarding FID scores, they are reported
> in Appendix A.10 Table 8 (the SAEMNESIA version reported in the main paper corresponds
> to OS-DC-CA-FT). The missing style metrics (Style UA, IRA, and CRA) are already
> reported in Appendix A.14. We report below the full comparison with SalUn included:
>
> | Method | Style UA (↑) | Style IRA (↑) | Style CRA (↑) | Object UA (↑) | Object IRA (↑) | Object CRA (↑) | Avg. (↑) | FID (↓) | Memory GB (↓) | Storage GB (↓) |
> |--------|-------------|--------------|--------------|--------------|---------------|---------------|---------|---------|--------------|---------------|
> | ESD | **98.58** | 80.97 | 93.96 | 92.15 | 55.78 | 44.23 | 77.61 | 65.55 | 17.8 | 4.3 |
> | FMN | 88.48 | 56.77 | 46.60 | 45.64 | 90.63 | 73.46 | 66.93 | 131.37 | 17.9 | 4.2 |
> | UCE | 98.40* | 60.22 | 47.71 | 94.31* | 39.35 | 34.67 | 62.45 | 182.01 | 5.1* | 1.7 |
> | CA | 60.82 | 96.01 | 92.70 | 46.67 | 90.11 | 81.97 | 78.05 | **54.21** | 10.1 | 4.2 |
> | SalUn | 86.26 | 90.39 | 95.08 | 86.91 | **96.35** | **99.59** | 92.43* | 61.05 | 30.8 | 4.0 |
> | SEOT | 56.90 | 94.68 | 84.31 | 23.25 | 95.57* | 82.71 | 72.91 | 62.38 | 7.34 | **0.0** |
> | SPM | 60.94 | 92.39 | 84.33 | 71.25 | 90.79 | 81.65 | 80.23 | 59.79 | 6.9 | **0.0** |
> | EDiff | 92.42 | 73.91 | 98.93 | 86.67 | 94.03 | 48.48 | 82.41 | 81.42 | 27.8 | 4.0 |
> | SHS | 95.84 | 80.42 | 43.27 | 80.73 | 81.15 | 67.99 | 74.90 | 119.34 | 31.2 | 4.0 |
> | SAeUron | 95.80 | **99.10** | **99.40** | 87.16 | 85.57 | 74.14 | 90.10 | 62.69 | **2.8** | 0.2* |
> | **SAEMNESIA (ours)** | 96.60 | 98.67* | 99.30* | **94.65** | 91.39 | 88.48* | **94.85** | 56.15* | **2.8** | 0.2* |
>
> *Second-best result.
>
> While SalUn achieves high Object CRA (99.59%), the reviewer's assessment was based on
> a single metric from the leaderboard. Looking at the full picture, SAEMNESIA achieves
> a higher overall average (94.85% vs. 92.43%) while requiring dramatically less memory
> (2.8 GB vs. 30.8 GB), an over 10× reduction that is critical for practical deployment.
> We will include this complete table in the revised version.
>
> **W2 - Unlearning of similar concepts (Dogs, Horses)**
>
> We acknowledge that visually similar concepts within the same semantic group present
> a greater challenge for concept erasure, and Dogs and Horses are indeed a known
> difficult pair due to their shared visual features (four-legged animals with similar
> body structure). This limitation is consistent with findings in SAeUron and is directly measured by our IRA metric, which captures within-domain
> retention accuracy.
>
> We note that the DC loss decorrelates activations across macro-groups (objects vs. styles) but not within them, which is the root cause of this interference. As discussed
> in our response to Reviewer q6GS W3, a promising direction to address this would be
> to incorporate pairwise semantic similarity into the supervised loss, applying stronger
> decorrelation penalties between highly similar concept pairs. We identify this as an
> important direction for future work.
>
> **Q1 - Experiments with more T2I models**
>
> We refer the reviewer to our response to W4 of Reviewer rGkx, where we address this point in detail and report preliminary SDXL results. In brief, SD v1.x remains the standard and widely-accepted backbone for concept erasure research, with the UnlearnCanvas benchmark itself built on SD v1.5. We acknowledge that validation on more recent architectures would strengthen the paper, and we have conducted preliminary experiments on SDXL leveraging the publicly available unsupervised SAE trained on SDXL Turbo by Surkov et al. [1] as initialization, with SAEMNESIA's supervised fine-tuning applied on top. SAEMNESIA substantially improves unlearning accuracy over the unsupervised SAE baseline (63.63% vs. 13.92%), while the trivial no-unlearning baseline averages 50.00%, confirming that the supervised signal generalizes to a different backbone. We plan to include these results in the final version.
>
> [1] Surkov et al., "One-Step is Enough: Sparse Autoencoders for Text-to-Image
> Diffusion Models." NeurIPS 2025.
>
> **Q2 - Hyperparameter optimization**
>
> We thank the reviewer for raising this point, as it was not sufficiently clear in the
> paper. The hyperparameters β, λ, and γ in Equations 6 and 7 are **not** optimized
> simultaneously with the model, they are selected via a grid search prior to training,
> using the validation loss (Equation 6) after 5 epochs as the selection criterion. The
> search covered the following values:
>
> - β ∈ {1.0, 3.0, 5.0}
> - λ ∈ {0.005, 0.010, 0.020}
> - γ ∈ {0.05, 0.10, 0.20}
>
> yielding 27 configurations in total. Once selected, these values are fixed and the
> model is trained normally. We will better clarify this procedure explicitly in the appendix of the camera-ready version.

---

> > ### Author Rebuttal · Reviewer_aQBR · 2026-04-03
> >
> > Thanks for the reply. Appreciate your explanation!

---

> > > ### Author Response · Authors · 2026-04-06
> > >
> > > We sincerely thank the reviewer for the thorough engagement with our rebuttal.
> > >
> > > We are glad our responses addressed the main concerns, and we will incorporate all the discussed improvements in the revision.

---

### Official Review · Reviewer_q6GS · 2026-03-08

**Soundness:** 2
**Presentation:** 3
**Significance:** 2
**Originality:** 2
**Overall Recommendation:** 4
**Confidence:** 3

**Summary:**

The paper introduces SAEMNESIA, a supervised Sparse Autoencoder framework for concept unlearning in text-to-image diffusion models. Compared to Concept Steerers, which operates on text encoders, and SAeUron, which trains SAEs in an unsupervised manner, the main idea of SAEMNESIA is to add supervised losses to SAE training that enforce one-to-one mappings between concepts and latent features, termed feature centralization. By binding each concept to a pre-specified feature, the method simplifies concept erasure to a single-latent intervention at inference time. The training takes place in two phases, i.e. unsupervised SAE training followed by supervised SAE training on in-domain data. On the UnlearnCanvas benchmark using Stable Diffusion v1.5, SAEMNESIA reports a 9.22% average improvement over the state-of-the-art SAeUron baseline, a 96.67% reduction in hyperparameter search, improved sequential unlearning scalability, stronger adversarial robustness, and competitive nudity removal on the I2P benchmark.

**Compliance With Llm Reviewing Policy:**

Affirmed.

**Final Justification:**

I thank the authors for their thorough rebuttal. The held-out concept evaluation addressed my concern about generalizability. I appreciate that the authors acknowledged the closed-vocabulary constraint and scalability as limitations to be discussed in the revision. I maintain my score.

**Key Questions For Authors:**

A few questions that would be interesting to address are the following:
- Q1. To check the generalizability mentioned in W1, can you provide a held-out concept evaluation where supervised training uses only a subset of concepts and erasure is evaluated on the remainder? Alternatively, could you provide results of a different evaluation dataset to measure how generalizability of the method in out-of-distribution scenarios?

- Q2. Regarding W2, if new concepts that need to be erased arise post-training, would you need to retrain the SAE from scratch or fall back to SAeUron? Therefore, what happens in the scenario that you need to erase a concept you have not identified during training?

- Q3. How does the SAE reconstruction fidelity change under SAEMNESIA compared to SAeUron? Specifically, how much reconstruction information is concentrated in the concept-assigned features versus the unassigned ones?

- Q4. The paper uses both TopK (k=32) and L1 regularization. Since TopK already enforces hard sparsity by construction, what is the distinct role of L1?

- Q5. With an expansion factor of 16 and k=32, the current setup maps 70 concepts into 20,480 latents with only 32 active per sample. How does the method scale when the number of concepts grows significantly?

In summary, I found the paper interesting. Welcoming feedback from the authors during the discussion period.

**Limitations:**

The authors acknowledge that extending SAEMNESIA to transformer-based diffusion architectures (e.g., FLUX) is future work. They also acknowledge the training and inference time trade-off, but this discussion needs to be included in the main paper. A limitation still to be addressed is the closed-vocabulary constraint.

**Strengths And Weaknesses:**

## Strengths

**S1: Feature Centralization.** The authors provide evidence that the supervised training achieves feature centralization, as observed in Figure 4 (and Figures 9-11 in Appendix A.6), showing a more concentrated score distribution compared to SAeUron's more dispersed one. The k-NN classification experiment in Figure 5 further strengthens this point.

**S2: Inference-time Efficiency gains.** SAEMNESIA is more efficient at test time compared to SAeUron. There is, however, a training-time trade-off, as SAEMNESIA requires approximately 5× the training time of SAeUron (Table 3, Appendix A.3). The authors have included it in the Appendix and would propose to highlight the trade-off in the main paper.

**S3: Sequential Unlearning.** SAEMNESIA achieves 92.4% unlearning accuracy for 9-object removal versus SAeUron's 64% (Figure 8), along with higher retention accuracy, representing a convincing practical advantage.

**S4: Ablations.** The authors have provided thorough ablations.

---

## Weaknesses

**W1: Supervision on Evaluation Concepts.** The CA loss trains each concept's assigned feature to highly activate when the concept is present. SAEMNESIA is therefore tested on the same concepts it was supervised on, which raises concerns about the generalizability of the evaluation.

**W2: Requirement of a Predefined Set of Concepts.** SAEMNESIA requires all erasure target concepts to be specified at training time. On the contrary, SAeUron can attempt novel concepts by recomputing scores without retraining. This asymmetry needs to be discussed since it is central to practical deployment.

**W3: Within-Group Interference.** The DC loss decorrelates across groups (objects vs. styles) but not within them. Figure 15 (Appendix A.11) confirms this: erasing Dogs drops Cats to 45.1%, erasing Human drops Statues to 0.0%. Semantically similar concepts within the same group still interfere despite the one-to-one mapping property

**W4: Feature Centralization Fragility.** The nudity experiment (Table 12, Appendix A.13) shows single-latent SAEMNESIA achieves 47 detections versus SAeUron's 18, requiring a two-latent fallback to reach 9 detections. Is there a principled criterion for when single-latent erasure is insufficient?

---

> ### Author Rebuttal · Authors · 2026-03-30
>
> **W1 / W2 / Q1 / Q2 - Closed-vocabulary constraint and generalizability**
>
> We acknowledge these are valid and related concerns, and we address them jointly as they all pertain to the closed-vocabulary nature of SAEMNESIA's supervised training.
>
> When a new concept needs to be erased post-training, there are two possible approaches. The first is to train a new SAE incorporating the novel concept, which yields the strongest concept-latent binding but requires additional training. The second, more practical approach is to compute the score function (Equation 4) over a small set of activations generated from prompts containing the novel concept, identify the highest-scoring latent, and use it directly for unlearning **without retraining the SAE**, analogously to how SAeUron handles novel concepts. This is precisely the inference-time procedure already used for concept-latent assignment, and it applies equally to unseen concepts.
>
> We followed SAeUron and conducted a held-out concept experiment where supervised training uses only half of the UnlearnCanvas styles and erasure is evaluated on the remainder, directly measuring generalization to unseen concepts:
>
> | Method | UA (↑) | IRA (↑) | CRA (↑) | Avg. (↑) |
> |--------|--------|---------|---------|---------|
> | SAeUron | 50.69 | 68.11 | 98.16 | 72.32 |
> | SAEMNESIA | **53.07** | **69.93** | **98.26** | **73.75** |
>
> SAEMNESIA consistently outperforms SAeUron across all metrics even on held-out concepts, demonstrating that supervised training generalizes beyond the concepts seen during training.
>
> Lastly, the risk of overfitting is further mitigated because training and evaluation use fundamentally different prompts: the structured format
> "An image of {object} in {style}" is used only during evaluation, while training uses semantically diverse prompts that do not explicitly name the target concept
> (e.g., "Colosseum", "Villa" for "Architectures"). See W2 of Reviewer rGkx for details.
>
> **W3 - Within-group interference**
>
> We agree this is a valid limitation. The DC loss decorrelates across macro-groups
> (objects vs. styles) but not within them. We note that this is an inherent challenge in any concept unlearning method, as also acknowledged by SAeUron. A promising direction for future work would be to incorporate pairwise concept similarity into the supervised loss: e.g., applying stronger decorrelation penalties between highly similar concept pairs via CLIP embeddings. We will include this discussion in the revision.
>
> **W4 - Feature centralization fragility**
>
> The weaker single-latent performance in the nudity experiment is directly attributable to limited supervision: the SAE is trained on only two prompts ("naked man" and "naked woman") vs. 80 prompts per concept in the object experiments. This setup was adopted deliberately to ensure fair comparison with SAeUron. Despite this, SAEMNESIA-TOP2 achieves 9 detections vs. SAeUron's 18, steering only **2 latents** compared to SAeUron's **205**, two orders of magnitude fewer. The number of available supervision prompts serves as a principled proxy for when single-latent erasure may be insufficient; TOP2 provides a natural fallback that still vastly outperforms unsupervised alternatives in intervention compactness.
>
> **Q3 - SAE reconstruction fidelity**
>
> SAEMNESIA achieves reconstruction loss 1.4349 and cross-entropy loss 0.0018, vs. SAeUron's 1.22. The modest reconstruction increase is expected given the CA and DC losses redirecting concept information into assigned latents. The negligible
> cross-entropy (0.0018) confirms this redirection succeeds. Figure 5 further shows that k-NN classification using only the top-scoring latent matches using *all* features, confirming concept-relevant information is concentrated in assigned latents rather than distributed across unassigned ones.
>
> **Q4 - Role of L1 regularization alongside TopK**
>
> TopK already enforces hard sparsity, but empirically L1 yields a consistent
> improvement: removing it drops average score from 91.51% to 89.96%. We hypothesize L1 provides softer gradient-based pressure on pre-TopK activations, encouraging more uniform suppression of non-concept latents rather than merely zeroing them post-hoc.
>
> **Q5 - Scalability with a growing number of concepts**
>
> We welcome further clarification during the discussion period. With 20,480 latents and 70 concepts mapped, only 0.34% of latents are concept-assigned, leaving substantial headroom. The k=32 constraint is per-sample: a single image typically activates at most 2 concept-assigned latents, leaving 30 slots for reconstruction. Our sequential unlearning results (Section 4.4) confirm scalability empirically: SAEMNESIA maintains 92.4% UA at 9 concepts vs. SAeUron's 64%. TopK is also not fundamental: prior work [1] shows ReLU SAEs can discover larger concept vocabularies, offering a natural scaling path.
>
> [1] Rao et al., "Discover-then-Name: Task-Agnostic Concept Bottlenecks via Automated Concept Discovery." ECCV 2024.

---

> > ### Author Rebuttal · Reviewer_q6GS · 2026-04-03
> >
> > I would like to thank the authors for their detailed responses.
> >
> > ---
> >
> > **W1 / W2 / Q1 / Q2 - Closed-vocabulary constraint and generalizability**
> >
> > I appreciate the additional experiments and clarifications. The held-out evaluation is helpful, and the fallback to SAeUron-style scoring for unseen concepts is a reasonable practical solution. I think the closed vocabulary should be discussed as a limitation in the revision.
> >
> > ---
> >
> > **Q5 - Scalability with a growing number of concepts**
> >
> > I would also highlight scalability as a limitation. While the method performs well in the current regime, the one-concept-per-latent design may become challenging in more realistic open-vocabulary settings involving a large number of concepts (e.g., 20K+ concepts). In such cases, both latent capacity and, more importantly, stable and disentangled concept–latent assignments may become bottlenecks, particularly given observed interference between similar concepts.

---

> > > ### Author Response · Authors · 2026-04-03
> > >
> > > We thank the reviewer for the helpful clarifications.
> > >
> > > Regarding W1/W2/Q1/Q2, we agree and will explicitly discuss the closed-vocabulary constraint as a stated limitation in the revision.
> > >
> > > Regarding Q5, we acknowledge that at very large concept vocabularies, both latent capacity and stable concept-latent disentanglement become potential bottlenecks. However, we note that SAEMNESIA already operates with 70 assigned concepts, which is comparable to or exceeding the scale of the most ambitious works in the literature: ScaPre [1], the most recent scalability-focused work, covers up to 50 concepts and explicitly markets this as a breakthrough, while MACE [2] represents the current absolute upper bound at 100 concepts, explicitly framed as "mass" erasure. Furthermore, the SAE capacity can be increased by scaling the expansion factor or latent dimension, providing a natural path to larger vocabularies. Beyond a certain scale, unlearning a very large fraction of the model's knowledge would make full retraining on a curated dataset (if possible) a more practical and principled solution than post-hoc erasure. We will acknowledge scalability as a limitation in the revision and identify hierarchical decorrelation (W3) as a promising direction for future work.
> > >
> > >
> > > [1] Deng et al., "Forget Many, Forget Right: Scalable and Precise Concept Unlearning in Diffusion Models." ICLR 2026.
> > >
> > > [2] Lu et al., "MACE: Mass Concept Erasure in Diffusion Models." CVPR 2024.

---

### Official Review · Reviewer_rGkx · 2026-03-11

**Soundness:** 3
**Presentation:** 3
**Significance:** 3
**Originality:** 2
**Overall Recommendation:** 4
**Confidence:** 5

**Summary:**

The paper addresses the problem of concept erasure in t2i diffusion models. The authors point out that existing sparse autoencoder (SAE)-based approaches often suffer from feature splitting, where a concept is distributed across multiple latent features, which reduces the efficiency and precision of concept removal. To address this issue, the paper proposes SAEMNESIA, a supervised SAE framework that conduct a one-to-one mapping between concepts and latent units, localizing each concept to a single latent feature.
Experiments on UnlearnCanvas, sequential multi-concept unlearning, adversarial prompts, and NSFW suppression show that the method improves concept removal while preserving unrelated generation quality.

**Compliance With Llm Reviewing Policy:**

Affirmed.

**Final Justification:**

The rebuttal adequately addressed my main concerns. I have raised my score from 3 to 4. The key contribution of SAEMNESIA lies in explicitly assigning concept-latent correspondences via CA and DC losses, enabling direct inference-time intervention without hyperparameter search.

**Key Questions For Authors:**

Please see weakness

I'd be happy to raise the score if most of the problems are resolved.

**Limitations:**

Yes

**Strengths And Weaknesses:**

**Strengths**

1. The paper proposes using a supervised SAE to establish correspondences between concepts and latent features, enabling a concept-to-single-latent mapping. The overall method design is reasonable and offers a degree of interpretability: the SAE is first trained to obtain sparse representations, supervised signals are then used to strengthen the binding between concepts and latents, and at inference time, concept removal is achieved by simply suppressing the corresponding latent.

2. The method performs inference-time feature manipulation, without modifying the original diffusion model parameters, making deployment relatively lightweight. In principle, the approach should also be straightforward to transfer across diffusion models with different backbones.

3. The experiments are reasonably comprehensive, with quantitative results showing improvements over existing baselines across multiple metrics. The paper also includes sequential multi-concept unlearning and adversarial robustness experiments, lending further support to the method's effectiveness under more challenging settings.

**Weaknesses**
1. Real diffusion representations are typically distributed and highly entangled, yet the core assumption of this paper is that each concept can be concentrated onto a single latent feature. The authors have not sufficiently demonstrated that such structure naturally exists within the model, so the method's effectiveness may to some extent rely on the relatively structured concepts found in benchmark datasets.

2. The supervised signals are derived directly from prompts such as "An image of {object} in {style}", from which concept–activation pairs are constructed. This setup is reasonable for structured datasets like UnlearnCanvas, but in more open-ended real-world scenarios, concept boundaries may be much fuzzier and clean labels may not always be available. It would be helpful to include ablation experiments to further assess the method's robustness.

3. The proposed method's design is formally a little similar to feature masking or latent suppression approaches at inference time. It would be beneficial for the authors to include a qualitative comparison with masking-based methods[1,2], in order to more clearly highlight the advantages of this approach in terms of representation learning or concept localization. You can also compare your method with more updated baseline [3,4] to further enhance the persuasiveness. Since time is limited, this does not necessarily require additional experiments and could be addressed through discussion and analysis.

4. The experiments are primarily conducted on earlier Stable Diffusion models (e.g., SD v1.x). Validation on more recent diffusion backbones (such as SDXL, SD3, or the Flux1.0) would help further demonstrate the method's applicability and generalization to modern diffusion architectures.

Given the time constraints, a large number of experiments are not required, but at least it should be possible to further demonstrate how to address the above concerns.

[1] Chavhan, Ruchika, Da Li, and Timothy Hospedales. "Conceptprune: Concept editing in diffusion models via skilled neuron pruning." arXiv preprint arXiv:2405.19237 (2024).

[2] Deng, Kaiyuan, et al. "Forget Many, Forget Right: Scalable and Precise Concept Unlearning in Diffusion Models." arXiv preprint arXiv:2601.06162 (2026).

[3] Li, Gen, et al. "Sculpting memory: Multi-concept forgetting in diffusion models via dynamic mask and concept-aware optimization." Proceedings of the IEEE/CVF International Conference on Computer Vision. 2025.

[4] Gong, Chao, et al. "Reliable and efficient concept erasure of text-to-image diffusion models." European Conference on Computer Vision

---

> ### Author Rebuttal · Authors · 2026-03-30
>
> **W1 - Assumption that concepts can be concentrated in a single latent**
>
> We acknowledge this is a valid theoretical concern. The assumption that concept-level structure can be compressed into a single latent feature is grounded in the **Linear Representation Hypothesis** [5,6,7], which posits that neural networks tend to encode semantic concepts as linear directions in activation space. If this hypothesis holds, SAEs provide a principled mechanism for surfacing and isolating these directions. Fig. 5 provides direct empirical support: k-NN classification using only the single top-scoring latent identified by SAEMNESIA achieves accuracy comparable to using *all* latent features, across all 20 object concepts. This strongly suggests that concept-relevant information is indeed concentrated rather than uniformly distributed, and that this structure is recoverable through supervised training rather than being an artifact of benchmark design.
>
> Furthermore, SAEMNESIA's effectiveness is demonstrated across three distinct evaluation settings: UnlearnCanvas (structured concepts), adversarial prompt robustness (UnlearnDiffAtk), and open-domain nudity removal (I2P), suggesting the method generalizes beyond structured benchmarks.
>
> [5] Park et al., "The linear representation hypothesis and the geometry of large
> language models." arXiv:2311.03658 (2023).
>
> [6] Elhage et al., "Toy models of superposition." arXiv:2209.10652 (2022).
>
> [7] Arora et al., "A latent variable model approach to PMI-based word embeddings." TACL (2016).
>
> **W2 - Robustness to open-ended prompts and label availability**
>
> We take the opportunity to correct a misleading description in the paper. The
> structured prompt format "An image of {object} in {style}" is used **only during
> evaluation** (e.g., to compute UA, IRA and CRA metrics in Table 1). Training uses
> the anchor prompts of the UnlearnCanvas benchmark, which are semantically diverse and do not explicitly name the target concept. For example, the "Architectures" concept is trained on prompts such as "Colosseum", "Villa", etc., rather than the literal label "Architectures" (full list [here](https://github.com/OPTML-Group/UnlearnCanvas/tree/920cfd14033521b9100ad5c80952c23d1f8f0393/machine_unlearning/mu_concept_ablation_ca/anchor_prompts)).
> The supervised signal is thus derived from **concept-level labels applied to semantically diverse prompts**, making the approach more robust to real-world concept fuzziness than the current description suggests. We will make this distinction explicit in the revision.
>
> **W3 - Comparison with masking-based methods and recent baselines**
>
> We will cite and discuss all four references in the revision. ConceptPrune [1] and
> ScaPre [2] permanently modify model weights — pruning FFN weights and applying closed-form weight updates respectively. Sculpting Memory [3] and RECE [4] similarly modify cross-attention weights. In contrast, SAEMNESIA operates **entirely at inference time without touching model weights**, making it reversible and transparent: it trains a separate SAE with explicit supervised concept-neuron assignments and steers activations at inference time, leaving the original model intact. Beyond this, weight-modification methods achieve erasure without explicitly modeling where concepts live, while SAEMNESIA learns an interpretable concept-latent structure grounded in the Linear Representation Hypothesis (see W1), representing a fundamental advantage in representation learning and concept localization. We will include this discussion in the related work section.
>
> **W4 - Validation on more recent diffusion backbones**
>
> We refer the reviewer to Section 4.1 for the full list of supporting citations. SD v1.x remains the standard backbone for concept erasure research: UnlearnCanvas is built on SD v1.5, and the majority of contemporary methods report results exclusively on SD v1.x. We acknowledge that extending to SDXL, SD3, or FLUX is non-trivial (Section 5), but report preliminary SDXL results using the unsupervised SAE from Surkov et al. [8] as initialization:
>
> | Method | UA (↑) | IRA (↑) | Avg. (↑) |
> |--------|--------|---------|---------|
> | No unlearning (trivial baseline) | 0.00 | 100.00 | 50.00 |
> | Unsupervised SAE [8] | 13.92 | 85.55 | 49.73 |
> | SAEMNESIA | **63.63** | 45.15 | **54.39** |
>
> The unsupervised SAE barely exceeds the trivial baseline (49.73%), while SAEMNESIA substantially improves UA (13.92% → 63.63%), confirming the supervised signal generalizes to a different backbone. The IRA drop and omission of CRA reflect the domain gap: UnlearnCanvas uses a fine-tuned SD v1.5 and ViT classifiers trained on SD v1.5 images, making direct metric comparison noisy. A fair SDXL evaluation requires dedicated retraining, which we identify as future work.
>
> [8] Surkov et al., "One-Step is Enough: Sparse Autoencoders for Text-to-Image
> Diffusion Models." NeurIPS 2025.

---

> > ### Author Rebuttal · Reviewer_rGkx · 2026-04-03
> >
> > Thank the authors for the thorough responses. The main concerns are largely addressed. W4 remain partially open but are acceptable given the time constraints. I have raised my score accordingly.

---

> > > ### Author Response · Authors · 2026-04-03
> > >
> > > We sincerely thank Reviewer rGkx for the thorough engagement with our rebuttal and for raising their score.
> > >
> > > We are glad our responses addressed the main concerns, and we will incorporate all the discussed improvements in the revision.

---

### Official Review · Reviewer_Q7pu · 2026-03-12

**Soundness:** 3
**Presentation:** 2
**Significance:** 2
**Originality:** 2
**Overall Recommendation:** 4
**Confidence:** 4

**Summary:**

The paper presents a supervised sparse autoencoder framework for concept erasure in diffusion models. Their method binds each concept to a single interpretable neuron such that disabling this neuron, prevents the generation of target concept without affecting others. Overall, they present strong erasure performance.

**Compliance With Llm Reviewing Policy:**

Affirmed.

**Final Justification:**

I believe all my concerns have been addressed.

**Key Questions For Authors:**

See Weaknesses.

**Limitations:**

The authors have highlighted limitations in their paper. The main limitation of this paper is that it cannot be generalized to multi-modal diffusion transformers like FLUX, Qwen-Image etc yet.

**Strengths And Weaknesses:**

**Strengths:**
1. The paper is well written.
2. The proposed method is technically sound.
3. The paper demonstrates strong erasure performance that outperforms previous baselines.

**Weaknesses:**
1. Lack of adversarial attacks - Thank you for including adversarial attacks in your paper. Can you also include some results for black box attacks like Ring-A-Bell and MMA-Diffusion to show that your method can generalize to other adversarial attacks as well?
2. Limited difference with SAEUron - I believe this work builds incrementally over SAEUron. Can the authors please highlight the differences in detail? From my understanding, SAEAmnesia adds a supervised component in the loss function compared to unsupervised-only SAEUron and enforces a one-to-one relation between concept and neuron, unlike many-to-many in SAEUron. I agree that it leads to better disentanglement and fewer neurons being 'shut down'. However, compared to SAEUron, technical novelty seems limited.
3. Missing computational analysis - Can you please add some analysis of the time required to erase one concept compared to SAEUron?
4. Multi-concept erasure - UCE and other concept erasure works demonstrate erasure of multiple artists at the same time from the model? Can your method be used for erasing multiple concepts simultaneously?
5. Does erasure of a particular concept unintentionally lead to erasure of a very similar concept? For example, Erasing 'dog' may affect the generation of cats. It has been seen that erasing Van Gogh style often affects Claude Monet's style. Does your method prevent this?

---

> ### Author Rebuttal · Authors · 2026-03-30
>
> **W1 - Black-box adv attacks**
>
> We thank the reviewer for this suggestion and provide Ring-A-Bell black-box adv attack results below. SAEMNESIA achieves an average Adversarial UA of **94%** compared to SAeUron's **78%**, demonstrating that the robustness gains
> observed under white-box attacks generalize to black-box settings.
>
> | Concept | Atk (no unlearn) | Atk (w/ unlearn) SAEMNESIA | Adv. UA SAEMNESIA | Atk (w/ unlearn) SAeUron | Adv. UA SAeUron |
> |---------|-----------------|---------------------------|------------------|--------------------------|----------------|
> | Bears | 0.9 | 0.2 | **0.8** | 0.3 | 0.7 |
> | Dogs | 1.0 | 0.0 | **1.0** | 0.2 | 0.8 |
> | Cats | 1.0 | 0.1 | 0.9 | 0.0 | **1.0** |
> | Horses | 1.0 | 0.0 | **1.0** | 0.6 | 0.4 |
> | Birds | 0.7 | 0.0 | **1.0** | 0.0 | **1.0** |
> | **Average** | | | **0.94** | | 0.78 |
>
> We were unable to include MMA-Diffusion results within the rebuttal deadline, but we will provide both analyses in the final version.
>
> **W2 - Limited difference with SAEUron**
>
> The supervised loss induces a **fundamentally different representational regime**. While SAeUron achieves *monosemanticity* (each latent activates for one concept), it does not achieve *feature centralization* (each concept in a single latent). SAEMNESIA enforces **both simultaneously**, enabling qualitatively distinct consequences:
>
> (i) Search complexity: SAeUron requires 210 hyperparameter evaluations per concept; SAEMNESIA requires only 7, a 96.67% reduction.
>
> (ii) Sequential scalability: SAeUron degrades to 64% at 9 concepts; SAEMNESIA maintains 92.4%. This degradation is **structurally unavoidable** under many-to-one mappings, no tuning can resolve it without explicit concept localization.
>
> (iii) Adversarial robustness: Effectiveness drop: 49.50% (SAeUron) vs. 40.10% (SAEMNESIA), a direct consequence of stronger concept-neuron binding.
>
> **W3 - Missing computational analysis**
>
> We respectfully point the reviewer to Appendix A.3, which is explicitly referenced in Section 4.2 of the main paper ("our method does not increase inference computational cost (see Appendix A.3)"), and already contains a detailed computational analysis. Inference overhead is reported as follows:
> | Metric | Without SAE | SAeUron | SAEMNESIA |
> |--------|------------|---------|-----------|
> | Mean time (s) | 3.550 ± 0.009 | 3.641 ± 0.011 | 3.617 ± 0.018 |
> | Overhead (%) | - | 2.56 | 1.89 |
>
> SAEMNESIA is in fact **faster than SAeUron at inference** (1.89% vs. 2.56% overhead), as the one-to-one concept-neuron mapping requires steering a single latent per concept rather than a combination of features.
>
> **W4 - Multi-concept erasure**
>
> We refer the reviewer to Section 4.4, where sequential unlearning is explicitly demonstrated. SAEMNESIA supports multi-concept erasure by cumulatively applying multipliers across concepts. We demonstrate this for up to 9 objects, achieving 92.4% unlearning accuracy compared to SAeUron's 64%, a 28.4% improvement. Retention accuracy also remains higher (60.9% vs. 48.4%), confirming that erasing multiple concepts does not degrade generation of unrelated content.
>
> We additionally report sequential unlearning results for styles, which further confirm the robustness of our approach. SAEMNESIA achieves slightly higher unlearning accuracy as the number of sequentially erased styles increases:
>
> | Method | Style 1 | Style 2 | Style 3 | Style 4 | Style 5 | Style 6 |
> |--------|---------|---------|---------|---------|---------|---------|
> | SAEMNESIA | 0.9684 | 0.9698 | 0.9905 | 0.9968 | 0.9928 | 0.9904 |
> | SAeUron | 0.9710 | 0.9718 | 0.9705 | 0.9710 | 0.9716 | 0.9722 |
>
> Notably, while both methods start at comparable performance, SAEMNESIA improves with accumulation whereas SAeUron remains flat, suggesting that our stronger concept-neuron binding becomes increasingly advantageous as more concepts are erased sequentially.
>
> **W5 - Unintended erasure of similar concepts**
>
> This is directly measured by our IRA and CRA metrics, reported in Table 1. IRA (In-domain Retain Accuracy) measures whether erasing a concept affects other concepts within the same domain (e.g., erasing Dogs affecting Cats), while CRA (Cross-domain Retain Accuracy) measures interference across domains (e.g., erasing an object affecting styles). SAEMNESIA outperforms SAeUron on both: **91.39% vs. 85.57% IRA** and **88.48% vs. 74.14% CRA**, confirming that our stronger concept-neuron binding reduces unintended interference compared to the baseline.
>
> We also provide a dedicated concept interference analysis in Appendix A.11 (Tables 10 and 11), specifically examining the Dogs vs. Cats case highlighted by the reviewer.
>
> Finally, we directly address the Van Gogh / Monet example raised by the reviewer. The following results show that erasing Van Gogh has zero impact on Monet, and vice versa:
>
> | Unlearned | Tested | UA (%) |
> |-----------|--------|--------|
> | Van Gogh | Van Gogh | 99.5 |
> | Van Gogh | Monet | 0.0 |
> | Monet | Van Gogh | 0.0 |
> | Monet | Monet | 100.0 |

---

> > ### Author Rebuttal · Reviewer_Q7pu · 2026-04-03
> >
> > Thank you for the rebuttal. My concerns have been addressed and I have revised my score accordingly.

---

> > > ### Author Response · Authors · 2026-04-06
> > >
> > > We sincerely thank the reviewer for the thorough engagement with our rebuttal and for raising the score.
> > >
> > > We are glad our responses addressed the main concerns, and we will incorporate all the discussed improvements in the revision.

---

### Decision · Program_Chairs · 2026-04-30

**Decision:**

Accept (regular)

**Comment:**

This paper investigates sparse autoencoders (SAE) for decoupling knowledge for machine unlearning, a direction that has been explored in prior work. Building on this framework, the authors propose a supervised SAE to achieve a one-to-one mapping between concepts and latent units. The reviewers agree that the paper is well written and demonstrates strong empirical performance.

At the same time, several weaknesses have been identified. In my view, the contribution is somewhat incremental, given the existence of multiple SAE-based approaches for machine unlearning. More importantly, as noted by one reviewer, real diffusion representations are typically distributed and highly entangled. This raises concerns about the practicality of building one-to-one mapping between concepts and latent units, especially in more realistic and complex scenarios. The current method is mainly evaluated on relatively simple, structured formulations (e.g., “an image of {object} in {style}”), and its generalization beyond such settings remains unclear.

Overall, the reviewers’ scores (4/4/4/3) are consistent and indicate a borderline but positive assessment. I agree that this work provides a meaningful improvement within a specific line of research (machine unlearning via SAE), although its broader impact may be limited.

Based on these considerations, I recommend weak accept.